# Nuclear magnetic resonance for wireless magnetic tracking

M. Efe Tiryaki [1,2,6], Pouria Esmaeili-dokht [1,3,6], Jelena Lazovic [1], Klaas P. Pruessmann [1,4] & Metin Sitti [1,5] ✉

Wireless trackers have emerged as a crucial technology in minimally invasive medical procedures with their remote localization capabilities. Existing trackers suffer from miniaturization issues and complex designs, which limit their integration into medical devices. We present nuclear magnetic resonance (NMR) magnetic sensing, a quantum sensing approach with nT sensitivity for wireless magnetic tracking. NMR magnetic sensing enables millimeter-scale tracking accuracy and versatile miniaturized tracker designs for minimally invasive medical devices in magnetic resonance imaging scanners. As examples, we demonstrate miniature magnetic trackers with submillimeter-scale diameters for guidewires and optic fibers, flexible magnetic trackers for soft devices, and ferrofluidic trackers for shape-morphing devices. With the demonstrated miniaturization and wide range of tracker design possibilities, wireless magnetic tracking with NMR is promising for future minimally invasive medical operations.

Recent developments in minimally invasive procedures and medical robotics have created a growing demand for medical device tracking methods in the human body, where a direct line of sight is not possible. Currently, medical device tracking is performed by medical imaging[1–6] and complementary remote sensing, such as electromagnetic (EM)[7–10] and magnetic sensing[11–14]. However, each approach has specific drawbacks in terms of spatial and temporal resolution, workspace, and miniaturization of tracking devices. The most widely used medical imaging methods, such as X-ray fluoroscopy and magnetic resonance imaging (MRI), suffer temporal resolution issues because of ionizing radiation exposure from X-ray or the inherent spatiotemporal resolution trade-off of MR imaging, respectively. Complementary remote sensing methods compensate for the temporal resolution issues of medical image-based tracking methods by introducing field generators[10,11] or onboard field sensors[12–14] as tracking devices. Among remote sensing approaches, commercial EM sensors[7] and onboard magnetic sensors[12–15] are the most common approaches. However, the tethered nature and relatively large size of these trackers limit their usage. Wireless EM tracking, including electrical or mechanical

resonators, addresses tether issues with a relatively small size, especially with recently proposed mechanical resonators[8–10]. Despite their small size, these wireless EM trackers limit overall medical device miniaturization due to the structural restrictions of resonator design. As an alternative, wireless magnetic tracking could enable further miniaturization and versatile tracker design by eliminating structural restrictions. However, the μT-level sensitivity of hall-effect magnetic sensors used in existing magnetic tracking systems is not sufficient for tracker miniaturization below the centimeter scale[11].

The required sensitivity could be achieved with quantum sensing approaches exploiting quantum properties of matter for magnetic field measurement[16], such as superconducting quantum devices[17], optically pumped magnetic sensors[18], electron spin magnetic resonance[19], and nuclear magnetic resonance (NMR) magnetic sensors[20]. While some of these quantum sensors have found their way into biomedical applications in magneto-encephalography and cardiography, they require high hardware and installation costs, preventing their mainstream adoption in clinical environments. Among these quantum sensors, NMR magnetic sensors offer a unique opportunity

[1]Physical Intelligence Department, Max Planck Institute for Intelligent Systems, Stuttgart, Germany. [2]Mechanical Engineering Department, Middle East Technical University, Ankara, Turkey. [3]Stuttgart Center for Simulation Science, University of Stuttgart, Stuttgart, Germany. [4]Institute for Biomedical Engineering, ETH Zurich and University of Zurich, Zurich, Switzerland. [5]School of Medicine and College of Engineering, Koç University, Istanbul, Turkey. [6]These authors contributed equally: M. Efe Tiryaki, Pouria Esmaeili-dokht. ✉e-mail: msitti@ku.edu.tr

to be integrated into the hardware of clinically available MRI scanners and provide inherent potential for intraoperative imaging. Moreover, operating at high magnetic fields, the NMR magnetic sensor enables the use of soft magnetic materials for trackers, which is not possible with other quantum sensors. This enables further miniaturization and design versatility due to higher saturation magnetization and simpler manufacturing processes compared to permanent magnetic trackers.

Considering the miniaturization and integration limitations of existing wireless tracking methods and the potential of NMR magnetic sensing, we developed a wireless magnetic tracking system using an NMR magnetic sensor array (Fig. 1). We localized miniaturized magnetic trackers, such as soft magnetic rigid, hollow trackers, flexible magnetic trackers, and ferrofluidic trackers, by remotely measuring the magnetic field inhomogeneity in an MRI scanner. The NMR magnetic sensor is composed of a small radiofrequency (RF) coil encircling a glass tube of water and a matching circuit (Fig. 2a). It measures the magnetic field difference,

$$\delta B = \gamma^{-1} \frac{\Delta \phi}{\Delta t} \tag{1}$$

relative to the background magnetic field ($\mathbf{B}_0 = B_0 \hat{\mathbf{z}}$) of the MRI scanner in $\hat{\mathbf{z}}$ direction of the MRI scanner through the rate of phase accumulation of nuclear spin precession in the free-induction decay (FID) signal of $^1$H nuclei in water placed in the sensor[20], where $\gamma$ is the gyromagnetic ratio of $^1$H nuclei (Supplementary Note 1 and Fig. 1). This nuclear spin frequency-modulated magnetic measurement provided nT magnetic field measurement sensitivity—three orders of magnitude lower than hall-effect sensors—and a broad magnetic measurement range of ±700 μT for 30 kHz excitation BW. The NMR sensors are integrated into a sensor array through an RF switch that connects to the RF hardware of the MRI scanner (Supplementary Fig. 2). This setup enables sequential excitation and acquisition of FID signals with a high temporal resolution of 25 Hz due to the short phase accumulation duration of 2.5 ms.

## Results

### Magnetic modelling at high fields for NMR magnetic tracking

Traditionally, magnetic tracking is performed using hard magnetic trackers, such as Neodymium magnets, due to their high remanence magnetization[11]. However, soft magnetic trackers, such as iron and spring steel, have higher saturation magnetization values at a high magnetic field (see Supplementary Note 2 and Supplementary Fig. 3); thus, they allow further miniaturization. Moreover, unlike hard magnetic trackers at low fields, the magnetization of soft magnets aligns substantially with the high $\mathbf{B}_0$ field[6], which decouples tracking from the tracker orientation (Supplementary Fig. 4). Consequently, we could model the measured high-field dipole magnetic field as (Supplementary Note 3):

$$B_d(\mathbf{r}) = \frac{\mu_0 M}{4\pi} \frac{3\|\hat{z} \cdot \mathbf{r}\|^2 - \|\mathbf{r}\|^2}{\|\mathbf{r}\|^5} \tag{2}$$

where $M$ is the saturation magnetization of the soft magnet, $\mathbf{r}$ is the relative position of the magnet to the sensor, $\hat{z}$ is the unit vector in the $\mathbf{B}_0$ direction, and $\|\cdot\|$ is the Euclidean norm. We verified the model by measuring the magnetic field of a 1 mm-diameter steel bead, 0.56 emu, along predetermined channels inside a 7-Tesla (7 T) preclinical small-animal MRI scanner (Supplementary Fig. 5). The magnetic field values varied between −7 and 14 μT at a 2 cm distance (Fig. 2b) and −70 to 110 nT at a 10 cm distance (Fig. 2c), with model matching between the 2–8 cm range. The observed deviation from the model at close distances, especially between 45° and 60°, is due to the curvature of the dipole field (Supplementary Note 4 and Supplementary Fig. 6). The deviation from the model at greater distances is due to the interaction of the magnetic tracker with the MRI's superconducting main field coils (Supplementary Note 5; Supplementary Figs. 7–8, and Supplementary Movie 1).

### Characterization of magnetic tracking workspace

Next, we built a hexagonally placed array of seven NMR magnetic sensors for magnetic tracking (Fig. 2d). Each sensor is placed in an RF

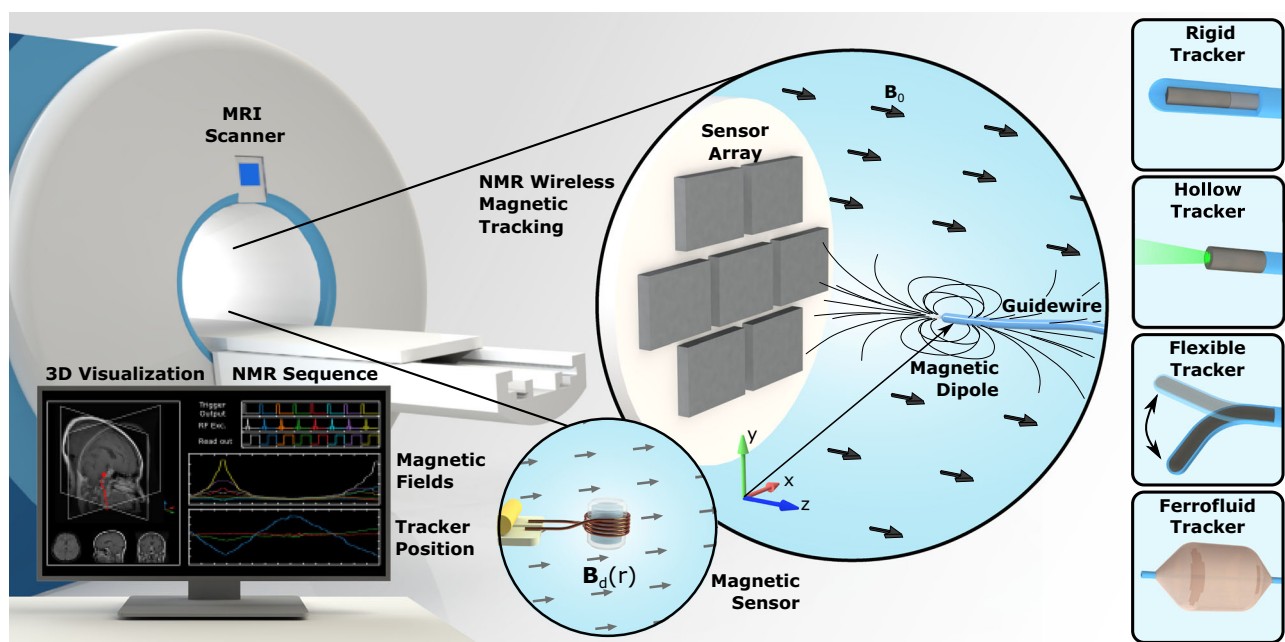

**Fig. 1 | Nuclear magnetic resonance (NMR) for wireless magnetic tracking concept.** The tracking concept, where the magnetic field of a magnetic tracker (e.g., a magnetic dipole placed at the tip of a guidewire) is measured by an array of seven NMR magnetic sensors. The 3D tracker position is shown in the user interface. Different magnetic tracker examples are depicted: rigid magnetic tracker, hollow magnetic tracker, flexible magnetic tracker, and ferrofluid magnetic tracker.

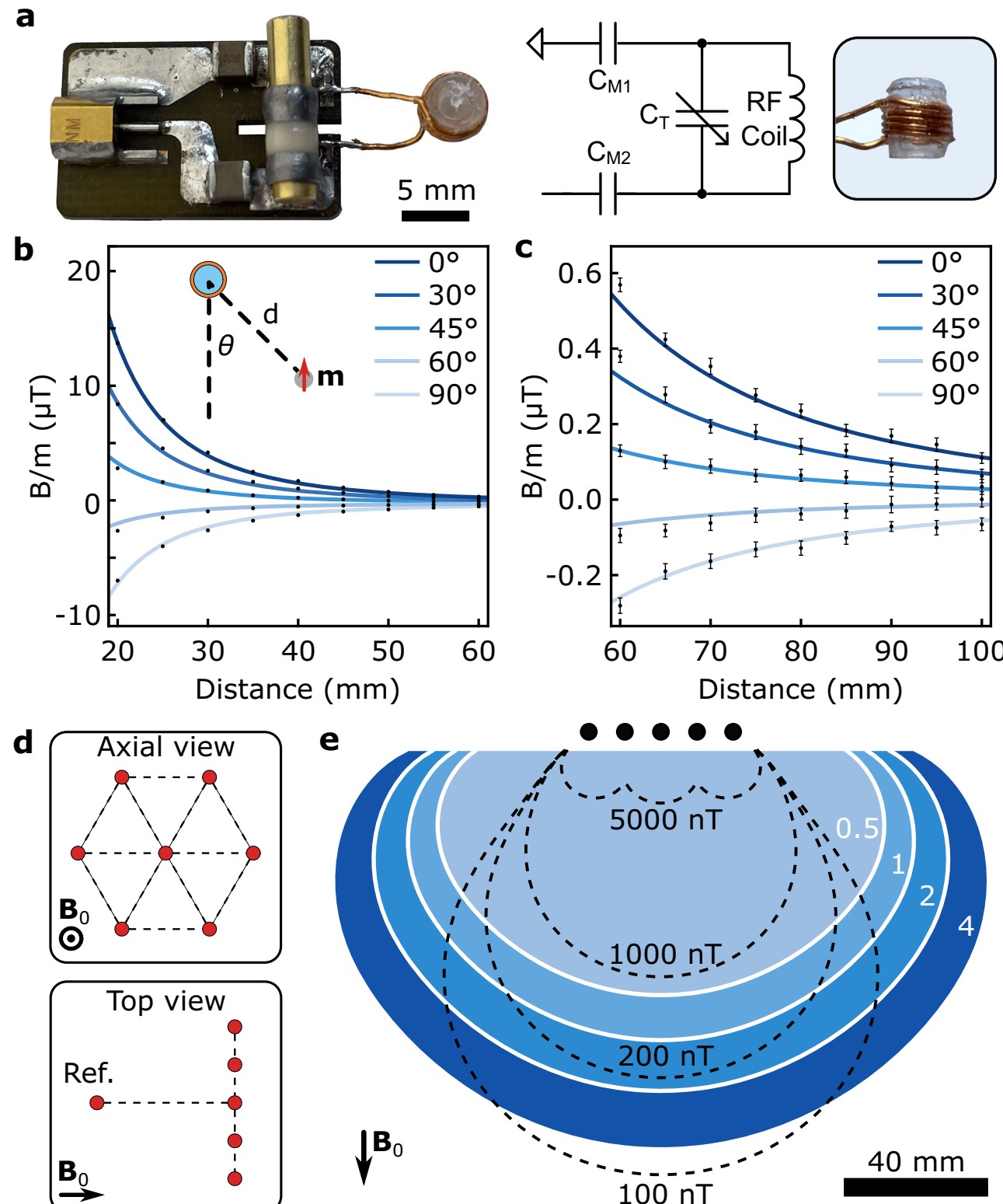

**Fig. 2 | NMR magnetic sensor array. a** The NMR magnetic sensor and the resonator circuit. **b**, **c** The magnetic field measurement of 0.65 emu magnetic dipole. The solid lines are the analytic model at different orientations. Points show the mean value and error bars represent the standard deviation (s.d.) of measured magnetic field. **d** The 3D configuration of the hexagonal sensor array and reference sensor in axial and top views. **e** The tracking workspace for 1 emu soft magnet. The dashed lines are the average magnetic field measured by seven sensors. The blue color map is the expected tracking precision.

shielding composed of aluminum foil (Supplementary Fig. 15a), which eliminates interference between sensors and also the nearby tissue (see Supplementary Note 13). We also introduced a reference NMR magnetic sensor placed far from the other sensors to monitor the

variation of the background field, which could change substantially over a long duration of operation (Supplementary Fig. 9). Then, we performed a gradient-based sensor position calibration with respect to the center of the MRI scanner using the gradient hardware of the

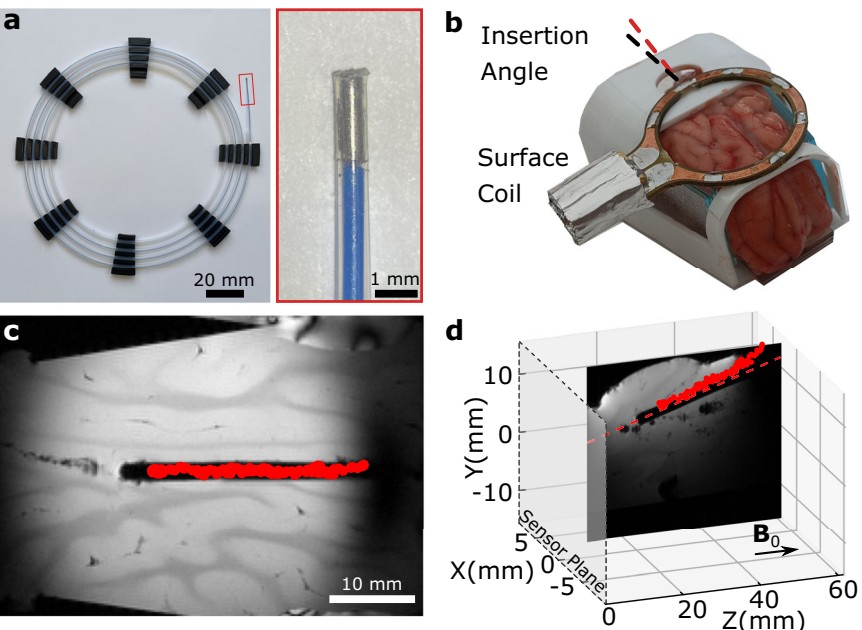

**Fig. 3 | Miniature rigid magnetic tracker. a** The magnetic guidewire has a 0.6 mm diameter and a spring steel tracker that is 2 mm (0.565 mm³). **b** The wireless magnetic tracking with NMR inside a porcine brain ex vivo. **c** The tracking data in the oblique plane MR image. The tracker positions are overlaid with red dots. **d** The 3D image tracking data overlayed on the MR image in the sagittal plane with respect to the sensor plane.

scanner (see "Methods", "sensor array calibration"). We investigated the theoretical limits for workspace for magnetic tracking with NMR using the soft magnetic dipole model Eq. (2). We model the lower bound for tracking precision for hexagonal planar sensor arrays using measurement covariance, $\mathbf{I} = \mathbf{H}^{\mathrm{T}}\mathbf{R}^{-1}\mathbf{H}$, where $\mathbf{H}$ and $\mathbf{R}$ are the local observation matrix and sensor noise covariance matrix. Then, we could express the static tracking precision as

$$\Sigma \propto \frac{\sigma_B \, r^5}{MV} \qquad (3)$$

where $\sigma_\mathrm{B}$, $r$, and $V$ are the sensor noise, the distance to the sensor plane, and the magnetic tracker volume, respectively. We calculated the modeled tracking precision for a 1 emu tracker in a 2D plane together with the mean magnetic field measured by tracking sensors (Fig. 2e), which showed a 1 mm lower bound of tracking precision in 100 mm proximity of the array.

Next, we performed tracking experiments on mechanically constrained linear channels for ground truth while tracking the guidewire with a particle filter-based tracking algorithm using magnetic field measurements (Supplementary Fig. 10). Then, we evaluated the precision and accuracy of the tracker position. Our tracking results showed less than 1 mm precision, matching our model, and below 2 mm accuracy in a $60 \times 80$ mm² workspace. We observed that the deviation from the dipole model prohibits tracking at a larger workspace in our small animal MRI scanner. However, the workspace could be further expanded in clinical MRI scanners. Increasing tracker size will increase workspace, according to Eq. (3), but it will reduce trackers' versatility and integration capabilities. Thus, we discuss methods to expand the workspace by modifying the sensors' position and number in the Supplementary Note 6 (Supplementary Fig. 11 and Supplementary Movie 2).

### Integration of miniaturized wireless magnetic trackers
To demonstrate tracker miniaturization, we developed guidewires with spring steel trackers at the distal end. The high saturation magnetization of steel, reaching $1.7 \times 10^6$ A/m², enables a magnetic moment

of 1 emu within 0.56 mm³, the smallest wireless tracker (Supplementary Table 1 and Supplementary Fig. 13). The small size allowed us to integrate the tracker in 0.8 mm-diameter guidewires (Fig. 3a). Trackers for smaller guidewires would also be possible with different aspect ratios (Supplementary Fig. 14). Later, we illustrated wireless tracking in an ex vivo porcine brain, simulating a needle insertion operation into the brain (Fig. 3b). We inserted a straight catheter in the brain and took pre-operational MR images of the brain using an imaging surface coil shown in Fig. 3c, d. Next, we manually switched the RF connection from the surface coil to the NMR sensor array. Then, we calibrated the background field with the diamagnetic brain tissue, which caused 1–2 μT variation on different sensors (Supplementary Fig. 15), and tracked the guidewire in the catheter in real time with NMR magnetic tracking (Supplementary Movies 3 and 4). We observed that the magnetic tracking matched with the catheter's image artifact with $0.78 \pm 0.30$ mm accuracy, where ±SD is a 95% confidence interval.

The main benefit of miniaturization with NMR magnetic sensing is the higher versatility in designing functional devices. While previous wireless tracking methods obstruct medical devices[10], we could design hollow structures to provide a working channel. To demonstrate the potential, we created a magnetically tracked laser fiber using a micromachined hollow iron tracker, smaller than 0.8 mm in diameter, at the distal end (Fig. 4a and Supplementary Movie 5). We showed that magnetic tracking could be performed in 3D space while the laser was continuously operated (Fig. 4b, c). We measured a tracking accuracy of $2.10 \pm 0.80$ mm with respect to the closest point in the center of the spiral channel, which is close to the 3 mm diameter of the channel (Supplementary Fig. 16). Next, we integrated the miniaturized tracker with 1.3 emu magnetic moment into a custom-made endoscope camera system, and performed tracking in ex vivo on a porcine esophagus to demonstrate wireless tracking while performing endoscope video capturing (see Supplementary Movie 6 and Supplementary Fig. 17).

### Deformable magnetic soft trackers
To further highlight the versatility of the tracker design, we developed flexible magnetic trackers made of magnetic soft composite materials (Fig. 5a). Soft magnetic microparticle-embedded elastomeric materials

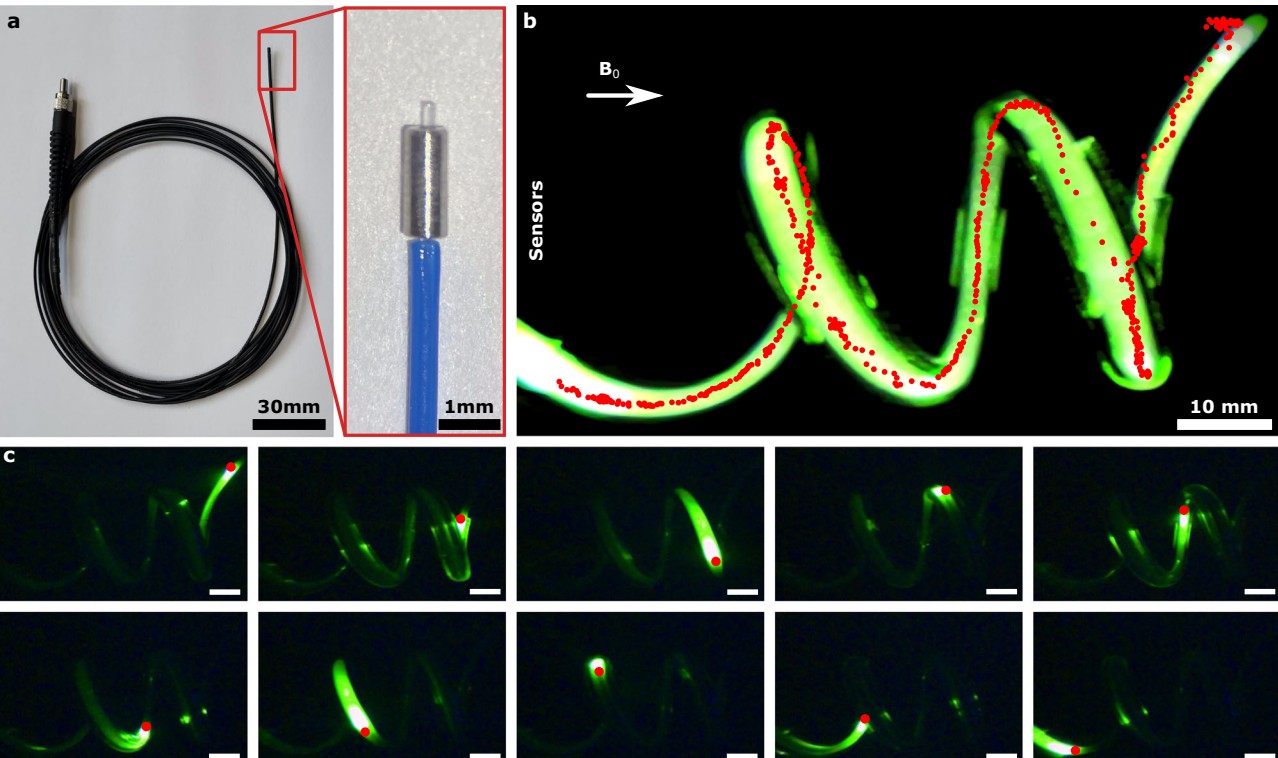

**Fig. 4 | Hollow magnetic tracker-integrated optic fiber tracking. a** The optic fiber with a laser connector. The hollow magnetic tracker with 0.8 mm outer and 0.3 mm diameter is placed at the tip of the optic fiber and fixed with glue and heat shrink tube. **b** The long exposure image of the optic fiber overlayed with the tracker position. **c** Snapshots of the experiment. The scale bars are 10 mm.

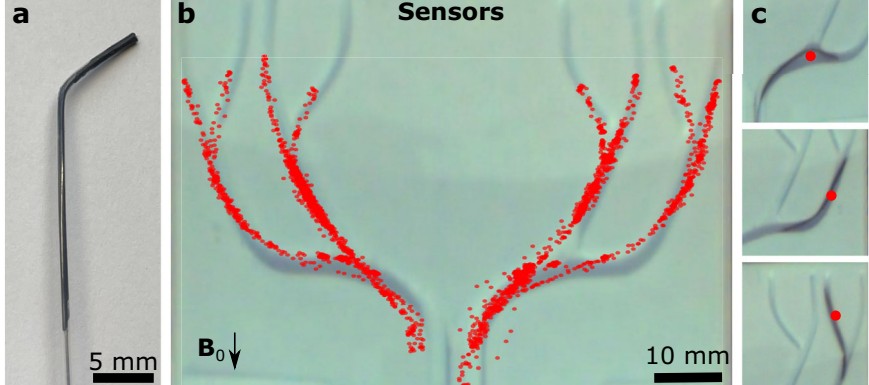

**Fig. 5 | Flexible magnetic tracker-integrated guidewire tracking. a** The custom-made angled-tip guidewire with a flexible magnetic tip. **b** The tracking data overlaid on the channels. The data is composed of four separate experiments targeting different channels each time. **c** The snapshots of the guidewire tracking during different shapes and positions of the flexible tip.

have recently emerged in miniaturized continuum robots[2]; however, the low magnetic moments of these soft magnets prevent their use in magnetic tracking with previous approaches. We created a curved guidewire with flexible magnetic trackers with 1.35 emus using a 3 μL 75% iron microparticle-silicone elastomer mixture at the distal end (Supplementary Fig. 18). To demonstrate functionality, we performed repeated manual navigation experiments inside 3D-printed channels (Fig. 5b and Supplementary Movie 7). We observed high-accuracy tracking of less than $0.72 \pm 0.67$ mm when the flexible tracker was in a straight configuration. However, the tracking error increased as high as 4.3 mm as the guidewire approached the sensor array (Fig. 5c). We discussed the shape effects of flexible magnetic trackers in the Supplementary Note 7 (Supplementary Fig. 19).

Finally, we illustrated the potential for ferrofluidic magnetic trackers using a balloon catheter filled with an iron-oxide ($Fe_2O_3$) nanoparticle solution (Fig. 6). Ferrofluids provide the potential for reconfigurable and shape-changing magnetic systems[21]. However, the low magnetization of ferrofluids has prohibited them from being tracked magnetically using previous magnetic tracking methods (Supplementary Fig. 20). First, we showed that the NMR magnetic sensing could be used to monitor the inflation of the balloon (Supplementary Fig. 21 and Supplementary Movie 8). Next, we demonstrated magnetic tracking of the ferrofluid-injected balloon with 1.75 mL volume and 0.25 emu magnetic moment (Supplementary Movie 9 and Fig. 6b). Although tracking precision is reduced due to low magnetic moment (Supplementary Fig. 22), the maximum tracking

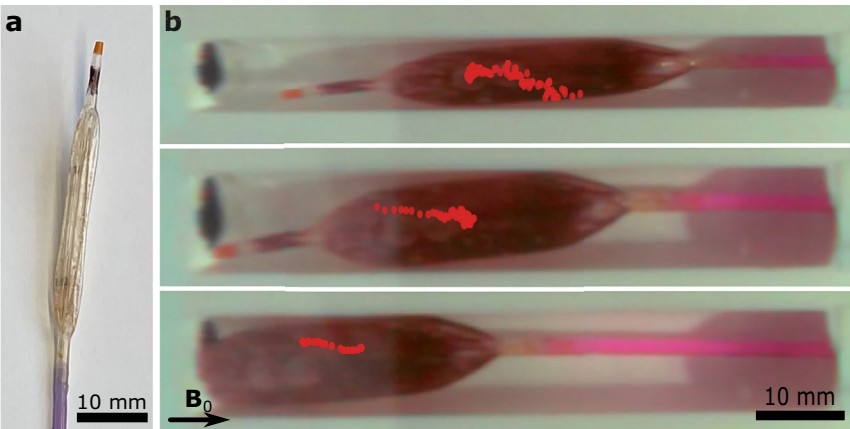

**Fig. 6 | Ferrofluidic magnetic tracker-based balloon catheter tracking. a** The empty commercial balloon catheter tip. **b** The snapshots of a balloon catheter inflated with the ferrofluidic tracker during the tracking experiment with the overlaid positions. The images are the initial point of corresponding overlaid tracking data.

error reaches 5.5 mm when the balloon approaches the sensors, remaining within the large volume of the balloon (Supplementary Note 8).

## Discussions

The magnetic tracking with NMR enables the miniaturization of magnetic trackers by ~3 orders of magnitude compared with previous magnetic tracking methods for the same workspace[11]. It provides a versatile tracker design for integration into functional medical devices. The distributed magnetic moment allows magnetic trackers to have diameters smaller than state-of-the-art wireless EM trackers[10]—ranging between 0.4–0.8 mm—which could be easily integrated into commercial guidewires with above 0.025"(-0.64 mm) diameter. The ability to track hollow magnetic trackers brings unparalleled versatility in tracker integration compared to other remote tracking methods. For instance, magnetic trackers could be placed around commercial catheters, biopsy needles, and neurostimulation electrodes without obstructing working channels. They could also be used together with sensing systems, such as micro-endoscope cameras (Supplementary Movie 3) and fiber Bragg grating sensors, for improved in situ biosensing and tissue interaction.

Moreover, previous remote tracking methods have been limited to rigid trackers, whereas most medical devices today are shifting to flexible and soft designs. For instance, an optimal tip stiffness profile is crucial for the effective and safe navigation of guidewires in vascular structures. The rigid remote trackers placed at the distal tip of guidewires prevent stiffness tuning and limit the steerability of the guidewire. The NMR magnetic sensing addresses this issue by allowing flexible magnetic trackers to be made using a soft magnetic powder and elastomer mixture, which enables magnetically trackable guidewires with flexible tips. These flexible magnetic trackers could also enable integration into emerging soft robotic designs[3]. At the other end of the rigidity spectrum, we can also track magnetic liquids composed of iron-oxide nanoparticles, which provide on-demand, injectable magnetic trackers. This could be used in marking certain tissues with magnetic liquid and tracking their motion when higher sensitivity NMR magnetic sensors are used in magnetic tracking[20].

Besides, NMR magnetic sensing provides a larger workspace compared to previous magnetic tracking methods[11] and a comparable workspace to wireless EM trackers of the same size[10], with similar tracking accuracy (Supplementary Table 1 and Supplementary Fig. 13), without the RF-induced heating risks. Although our demonstrations have been limited by the size of our preclinical MRI scanner, the NMR magnetic sensing technology offers a potentially larger workspace due to the relatively small size of NMR magnetic sensors compared with

planar coils used in EM trackers[10]. It is possible to create denser NMR magnetic arrays with more sensors or distribute the sensors around a desired volume to increase the workspace without interference from one another. Increasing sensor numbers could also be used to increase tracking accuracy further; however, this will limit the temporal resolution of tracking and require parallel hardware instead of the current serial hardware. Importantly, NMR magnetic sensing also addresses the position-calibration challenges that EM tracking systems face during integration with medical imaging. The gradient-based sensor positioning enables spatial calibration without the need for external hardware, allowing the magnetic tracker to be registered directly onto MR images (Fig. 2f) and enabling single-stage, real-time positioning of stereotactic neurosurgery robots[4]. Finally, NMR trackers do not suffer from the dead-angle issues inherent to EM trackers, since NMR tracking does not require external excitation[9].

Additionally, the NMR magnetic sensing technology addresses certain drawbacks of MR image-based tracking. Despite continuous improvements in MRI speed, real-time 3D tracking with passive fiducial markers, such as magnetic[22], 19-fluorine[23], and RF coils[24], and active MRI pickup coils[25], remains limited by slice thickness[22] and RF-induced tissue heating. In contrast, NMR sensors provide continuous 3D tracking without requiring the deposition of RF energy (see Supplementary Note 14), enabling continuous navigation with no risk of heating. Furthermore, NMR magnetic sensing eliminates acoustic noise, which in fast MRI-based tracking can exceed safe exposure limits[3,5] and hinder communication between clinicians during interventional MRI operations. Because NMR does not rely on MR gradients (Supplementary Note 10; Supplementary Fig. 23 and Supplementary Movie 10), it is inherently silent, creating a safer acoustic environment for both patients and clinicians. Moreover, NMR tracking also functions in air-filled cavities, where most MRI-based tracking methods, except $^{19}$F tracking[23], fail due to the absence of $^1$H of the surrounding water or tissue, allowing for the uninterrupted tracking of tools such as needles during insertion from outside the skin.

We envision that the NMR magnetic sensing could also be used in conjunction with 2D MRI and recently emerging MRI-powered magnetic actuation methods, by using an alternating MR sequence[5]. We can perform high-speed magnetic tracking with NMR simultaneously during slower MR image acquisition, which is safe in terms of RF heating and acoustic noise, and we can also integrate magnetic actuation at the same time. Furthermore, while clinical use of high-field MRI scanners is becoming popular[26], the integration of NMR magnetic sensor arrays into lower field MRI scanners, such as 1.5 T and 3 T, and emerging ultra-low-field MRI scanners could increase the affordability of magnetically guided minimally invasive operations in the future[27]; However, we must note that the magnetic sensing precision will

decrease with the static field strength since signal-to-noise ratio in MRI scanners scale with $\propto B_0^{7/4}$[28]. Moreover, NMR magnetic sensors constructed in a single-sided MR configuration could allow the use of magnetic tracking outside of the MRI scanners[29].

There are also certain limitations of magnetic tracking with NMR. First, it cannot be used on patients with ferromagnetic implants, a general limitation inherent to MRI technology. The second limitation is the difficulty in tracking magnetic trackers from near-moving tissues. While the magnetization calibration enables tracking near static tissues by subtracting the diamagnetic background, tracking near moving tissues, such as the chest during breathing motion, is challenging due to changing background signals[20]. This issue could be addressed in the future by incorporating the actuation dynamics into the particle filter rather than using a static model and modifying the proposed background field estimation using basis functions that approximate the dynamic component of the background[30], while utilizing more NMR magnetic sensors for increased measurement redundancy. Another limitation is the number of trackers that can be tracked accurately. While two magnetic trackers could be tracked with the current sensor number, seven, the tracking accuracy decreases due to increased measurement covariance (Supplementary Note 11; Supplementary Figs. 24 and 25). The sensor number should be increased, and sensors should be distributed over the tracking space to use a higher number of trackers.

In conclusion, NMR for wireless magnetic tracking enables versatile miniature wireless magnetic trackers inside MRI scanners. Demonstrating a high-sensitivity quantum sensing approach for magnetic tracking, the presented wireless magnetic tracking approach paves the way for precise, real-time localization for minimally invasive medical operations and medical robotic instruments.

## Methods

### Magnetic sensor
The NMR sensor comprises a 5-turn inductive coil of 400 μm diameter copper wires wrapped around a cylindrical glass capillary with an outer diameter of 4 mm and a height of 5 mm (Fig. 2a). After the winding, the capillaries are filled with deionized water and sealed permanently with epoxy (Loctite 401) to prevent evaporation. We use two constant matching capacitors, ranging from 0.5 to 1 pF, and one 0.5 to 12 pF range tuning capacitor (Knowles Voltronics) to tune the frequency of the resonator circuit to 300 MHz and match it to a 50 Ω impedance. The sensor is placed inside an aluminum-shielded box to prevent crosstalk between neighboring sensors and reduce the effects of high-frequency noises. A capacitive balanced network is used to tune and match the coils through an iterative process of tuning and adjusting the matching capacitors while connected to a Network Analyzer (Keysight E5061B). The matching capacitors are manually selected for each RF coil.

### Sensor switch
The switching setup features a non-reflective RF switch (HMC253ALC4, Hittite Microwave Corporation, USA) and a microprocessor (Arduino Uno) (Supplementary Fig. 2). The Arduino is connected to the trigger-out channel of the 7 T preclinical MRI scanner (Biospec 30/70, Bruker, Germany) and three input channels of the RF switch using optocouplers (FOD8001, Onsemi) to prevent any noise originating in external systems. A Li-ion battery, combined with an ultra-high PSRR (power supply rejection ratio) voltage regulator (LT3097, Analog Devices Inc.), powers up the RF switch. Except for the RF switching board, all the other components of the system were placed outside of the MRI scanner. The connections with the RF switch are made through non-magnetic BNC cables (mmcx connectors, Clinch Connectivity Solutions Johnson). The Arduino code sweeps eight channels in sequence by changing channels in response to a trigger signal from an MRI scanner.

### NMR sequence
The NMR sequence for magnetic field measurement is composed of basic free induction decay (FID) measurements and a trigger-out signal for synchronization with the sensor switch (Supplementary Fig. 2). The trigger-out allows Arduino to switch among the different sensors while the NMR sequence operates, without explicitly knowing which sensor is being measured. To synchronize the Arduino and the sequence, we reset the sensor counter in the Arduino code if an NMR signal is not received for more than 2 s. In the NMR sequence, we used an excitation pulse with a sinc shape, 0.2 ms RF duration ($T_{RF}$), 31.05 kHz transmission bandwidth (BW), 32 μW RF power, and a readout with 2.5 ms acquisition time, 250 sampling, 100 kHz receiver BW. The repetition time for a single sensor is 5 ms, and for an eight-sensor array, it is 40 ms. The parameters are used in all magnetic sensing experiments.

### Magnetic field calculation
The phase of the FID signal is calculated using the real and imaginary parts provided by the scanner, and phase unwrapping is applied (Supplementary Fig. 1a, b). The first 50 phase data points (0.5 ms) of the phase signal are discarded due to the settling time of the receiver and electronics in the readout system. Additionally, the inhomogeneous B1 field, accompanied by the attenuating nature of the FID signal, introduces nonlinear phase accumulation over time until the total noise dominates the signal amplitude (Supplementary Fig. 1b). To quantify this effect, we calculated SNR by comparing the maximum of the filtered FFT with the amplitude of deviation from the curve as a function of acquisition time (Supplementary Fig. 1c); thus, only the following 200 data points are used for further processing. The phase slope is calculated using linear fitting, and the magnetic field is calculated using Eq. (1). We used Eigen and FFTW libraries in C++ for real-time signal processing.

### Sensor array calibration
The calibration routine is composed of background field and sensor position calibrations. The background field $\Delta B$ varies with shimming configuration, the patient's tissue, and the main field coil temperature. Therefore, we need to perform a $\Delta B$ calibration. In stationary conditions, we collect 10 s of magnetic field measurements with all sensors to calculate the mean ($\mu_{\Delta B}$) and standard deviation ($\sigma$) of the $\Delta B$ individually for each sensor (Supplementary Fig. 15b, c). We subtract $\mu_{\Delta B}$ from the sensor reading during operation and use $\sigma$ in the particle filter for sensor noise.

Next, we perform sensor position calibration using the gradient hardware of the MRI scanner. We apply known gradient values ($G$) in positive and negative directions in the x-y-z axes for 5 s in positive and negative directions and record magnetic field measurements from all sensors (Supplementary Fig. 15d, e). Later, the average field in each direction is calculated for each sensor, and then we calculate each sensor's position using

$$\mathbf{r}_{s_i} = \frac{1}{2}\left(\frac{B_{s_i}^+ + B_{s_i}^-}{G}\right) \tag{4}$$

where $\mathbf{r}_{s_i}$ is the position of the $i$th sensor relative to the center of the MRI scanner, $B_{s_i}^+$ and $B_{s_i}^-$ are the measured average magnetic fields in positive and negative directions. To cancel the sensor geometry imperfection error, we averaged the measured value in opposite directions to calculate the position in each direction.

As the positions of sensors are a crucial factor in tracking precision, we designed an experiment to determine calibration accuracy in various conditions (Supplementary Fig. 15d–i). The calibration process was repeated 10 times for the system with seven sensors, both with and without the patient's stage, and with a porcine brain sample on the stage. Under all conditions, we achieved an error of less than 5 μm in the x and z directions and less than 10 μm in the y direction. Due to the

sensors' asymmetric profile in the $y$ direction, we observed more errors. Moreover, we observed an average sensor noise of 8 nT in all experiments, with no notable difference.

## Dipole measurement

A custom stage composed of different channel structures and known sensor positions was 3D printed using PLA material to validate the dipole model (Supplementary Fig. 5). Later, sensor heads were positioned inside the designated locations and connected to a matching circuit. This method, combined with the triangled 3-sensor setup, ensured the precision of the position and angle of the sensor compared to the channels and $B_0$ field, respectively. The 1 mm-diameter steel bead magnetic tracker was moved manually with 5 mm incremental steps during the experiment using a thin guidewire with delays of 3 s in each step.

## Measurement covariance

To calculate the theoretical lower bound for the sensing precision of a sensor set, we used the local observation matrix, $\mathbf{H}(x_i, y_i, z_i)$, by linearizing the sensor function, $\mathbf{h}(\mathbf{r}) = [\mathbf{h}_1, \mathbf{h}_2, \mathbf{h}_3, ..., \mathbf{h}_7] \in \mathbb{R}^7$, where $\mathbf{r}$ is the position of the magnet in the inertial frame, $\mathbf{h}_i(x_i, y_i, z_i)$ is the sensing function for ith sensor

$$\mathbf{h}_i(x_i, y_i, z_i) = \frac{\mu_0}{4\pi} m \frac{2z_i^2 - x_i^2 - y_i^2}{(x_i^2 + y_i^2 + z_i^2)^{\frac{5}{2}}} \tag{5}$$

and $x_i$, $y_i$, and $z_i$ are the distances between the magnet and the sensor in Eq. (2). The local observation matrix in the form of the Jacobian matrix, $\mathbf{H} \in \mathbb{R}^{3 \times 3}$, is calculated by

$$\mathbf{H} = \frac{\partial \mathbf{h}}{\partial \mathbf{r}} \tag{6}$$

The linearized static sensor precision, $\mathbf{I} \in \mathbb{R}^{3 \times 3}$, is calculated as

$$\mathbf{I} = \mathbf{H}^T \mathbf{R}^{-1} \mathbf{H} \tag{7}$$

where $\mathbf{R} = \mathrm{diag}([\sigma_1, \sigma_2, \sigma_3, ..., \sigma_6]) \in \mathbb{R}^{7 \times 7}$ is diagonal sensor covariance matrix. The diagonal elements of $\mathbf{I}$ give the precision in $\sigma_x$, $\sigma_y$, and $\sigma_z$ directions. We calculated the final precision by

$$\Sigma = \sqrt{\sigma_x^2 + \sigma_y^2 + \sigma_z^2} \tag{8}$$

## Particle filter

A particle filter is a recursive Bayesian state estimation technique that uses the Monte Carlo method to approximate the posterior distribution of the estimated state in three steps: propagation, reweighting, and resampling[31]. It has been used in the localization of dipole-based EM trackers[32]. The algorithm first approximates the state distribution using $N$ particles, which are distributed uniformly over the state space. Then, in the propagation step, it predicts the next time evolution of each particle using a prediction model with noise. Next, in the reweighting step, it approximates the posterior distribution of the state by calculating the probability of each particle contributing to the observed sensor measurement using a measurement model with noise. Finally, in the resampling step, the algorithm samples $N$ new particles from the approximated posterior distribution and replaces the previous set of $N$ particles with new particles. Then, the algorithm recursively repeats itself for each time step with new sensor data. This method recursively refines the estimated position using the obtained sensor data, generally converging within 1–2 time steps if $N$ is sufficiently large. To ensure convergence in the first step, a large initial particle number, $N_0$, is used. Then, in the next step, the particle number is reduced to $N$ during the resampling step.

To estimate the magnetic tracker position, we employed a particle filter-based state estimation model with the state, $\mathbf{x} = [\mathbf{r}^T, \Delta B] \in \mathbb{R}^4$, composed of tracker position and monitored background magnetic field variation. We consider the tracking problem as a static prediction model and a nonlinear measurement model:

$$\mathbf{x}(k) = \mathbf{x}(k-1) + \mathbf{v}(k-1) \tag{9}$$

$$\mathbf{z}(k) = \mathbf{B}_d(k) + \mathbf{\Delta B}(k) + \boldsymbol{\omega}(k) \tag{10}$$

where $k$ is the time step of estimation, $\mathbf{z}$ is the measurement vector $\in \mathbb{R}^8$, and $\mathbf{v}$ and $\boldsymbol{\omega}$ are the prediction and measurement noise modeled as a Gaussian distribution with covariance $\boldsymbol{\Sigma}_v = \mathrm{diag}([\sigma_x, \sigma_y, \sigma_z, \sigma_{\Delta B}]) \in \mathbb{R}^{4 \times 4}$ and $\boldsymbol{\Sigma}_\omega = \mathrm{diag}([\sigma_{s0}, \sigma_{s1}, ..., \sigma_{s7}]) \in \mathbb{R}^{8 \times 8}$. We used the prediction noise with $\sigma_x = \sigma_y = \sigma_z = 1$ mm and $\sigma_{\Delta B} = 0.1$ nT. The measurement noise is calculated during the sensor array calibration. $\mathbf{B}_d(k) = [B_d(\mathbf{r}_0), B_d(\mathbf{r}_1), ..., B_d(\mathbf{r}_7)]^T \in \mathbb{R}^8$ is the vector of the sensor measurement model of the sensors at time step k. Note that the reference sensor is included in the estimation as the 0th sensor to estimate the change in the background field.

We started the particle filter with $N_0 = 20,000$ and reduced the particle number to $N = 500$ after the first iteration of estimation. First, the algorithm propagates the particle position and background predictions using Eq. (8). Next, we calculate the expected sensor measurement, $\bar{\mathbf{z}}$, for each particle using Eq. (9) with the noise term. Then, we calculate the measurement likelihood for each particle using a normal distribution,

$$P_i(\bar{\mathbf{z}}_i | \mathbf{z}) = N(\bar{\mathbf{z}}_i | \mathbf{z}, \boldsymbol{\Sigma}_\omega) \tag{11}$$

where $i$ is the index of the particle in $N$ particles. Later, we scale the particle probabilities, $\beta_i = \frac{P_i}{\sum_0^N P_0}$, between 0 and 1. Finally, we resampled the particles using a random number $r$ from a uniform distribution (0, 1). To avoid prediction errors due to the inhomogeneity effect in proximity to the sensors, if any sensor measurement exceeds 20 μT, we remove the measurement of the corresponding sensor, exclude it from the resampling stage of the particle filter, and estimate the position using the remaining sensors.

## Workspace experiment

To validate the precision and accuracy of the tracking in the simulated workspace, a custom stage with parallel channel structures was 3d printed using PLA material (Supplementary Fig. 10a). The stage was mechanically fixed inside the MRI bore, and the channels were extended to the outside of the MRI using PTFE tubes. During the experiment, after calibration, the magnetic tracker, composed of two 1 mm-diameter steel beads, was placed inside one channel and moved manually in 5 mm incremental steps using a thin guidewire, with a 3-s delay between each step. Finally, the positions were estimated 10 times for each channel using the particle filter (Supplementary Fig. 10b, c).

## Guidewire and laser manufacturing

During the guidewire preparation, we used an MRI-compatible 0.5 mm-diameter glass optic fiber as the elastic core and 0.8 mm-diameter Teflon tubes with a 0.1 mm wall thickness (Adtech, UK). We glued a glass fiber, a Teflon tube, and the 0.6 mm-diameter 2 mm-height spring steel magnetic tracker. We prepared the laser by removing the coating from the last 2 mm of the optical fiber. Then, we glue a micromachined iron tracker with an outer diameter of 0.8 mm, an inner diameter of 0.3 mm, and a length of 2 mm. Later, we assembled the optical fiber with a laser connector and placed a black heat-shrink tube to prevent light leakage. We prepared the soft guidewire with a flexible magnetic tracker using a 0.8 mm-diameter Teflon tube and 0.5 mm-diameter optic fiber as the core. At the last 20 cm of the distal

end, we replaced the optic fiber with a 0.3 mm-diameter tungsten wire with an angled tip. We filled the tip with 3 μL of soft magnetic material, comprising a 75% mass ratio of μm-sized iron powder and Ecoflex. The endoscope system comprises a 1 mm × ~1 mm miniature camera (ams NanEye2D) inside a 4 mm diameter Teflon tube (Adtech, UK) and two 1 mm steel beads as the magnetic tracker. The tracker is put at the back of the camera and both of them are fixed with epoxy inside the tip of the Teflon tube. Finally, the camera is connected through a capture box (ams NANO-FIB-BOX) to save the video feed to computer.

## Data availability
The raw data supporting the findings of this study are available in the Edmond repository at https://doi.org/10.17617/3.XZAQ4Z.

## Code availability
The analysis code supporting the findings of this study are available in the Edmond repository at https://doi.org/10.17617/3.XZAQ4Z.

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

## Acknowledgements

The brain image inside Fig. 1 is from the Brain Tumor MRI Dataset, Kaggle under CC0. Materials from biorender.com were used for Supplementary Figs. 12a, d, f, g, i, 17a, and 28a. We thank Sinan Ozgun Demir, Gaurav Gardi and Asli Aydin for the theoretical discussions, and Devin Sheehan for help with the ex vivo experiments. The authors thank the International Max Planck Research School for Intelligent Systems (IMPRS-IS) for supporting Pouria Esmaeili-Dokht. Funding: This work was funded by the Max Planck Society and European Research Council Advanced Grant SoMMoR project (grant no. 834531).

## Author contributions

Conceptualization: M.E.T., P.E., M.S. and K.P.P., Methodology: M.E.T. and P.E., Software: M.E.T., P.E and J.L., Formal analysis: M.E.T. and P.E., Investigation: M.E.T. and P.E., Visualization: M.E.T. and P.E., Funding acquisition: M.S., Supervision: K.P.P. and M.S., Writing–original draft: M.E.T., P.E., M.S. and K.P.P.

## Funding

## Competing interests

The authors declare no competing interests.
