## [Transparent Peer Review File · Nature Communications]

Nuclear Magnetic Resonance For Wireless Magnetic Tracking

Corresponding Author: Professor Metin Sitti

Version 0:

Reviewer comments:

Reviewer #1

(Remarks to the Author)

Tiryaki et al. describe a novel way to perform magnetic tracking using a soft ferromagnetic marker and an array of NMR probes in an MRI instrument. I'll start by saying this is probably not the way I would go about tracking using an MRI, but it is an interesting way to do it and might have its advantages. The authors have done a solid job in their performance analysis and their consideration of many different effects, so I think it will be publishable in Nature Communications, but I do have some questions, concerns, and suggestions below that need to be addressed.

1. I'm not completely clear about the use case for this device. Is this for identifying the location of a catheter during medical procedures, where one wants to overlay the location with an MRI image? Or is it more general and just using the MRI as a big magnet? One of the cited papers was tracking bees, for instance. If it is medical uses, there is a whole field of catheter tracking that has been ignored in the introduction. There are even commercially sold devices that do this for MRI with similar accuracy to the manuscript over a larger volume. I provide a short list here:

<https://ajronline.org/doi/10.2214/ajr.183.2.1830391>
<https://onlinelibrary.wiley.com/doi/10.1002/mrm.24654>
<https://onlinelibrary.wiley.com/doi/full/10.1002/mrm.20202>
<https://inventions.techventures.columbia.edu/technologies/wireless-magnetic--1438>
<https://www.robinmedical.com/endoscout.html>

I'd like the authors to better describe the advantages and weaknesses of their technique in relation to these, because their method is quite different, but the application is not unique.

2. If the use case is to use the device in conjunction with MRI, there needs to be some discussion and analysis of how well the techniques can coexist as well as the safety implications. Ferromagnetic materials are usually considered incompatible with MRI for a number reasons. A) They distort the image, so they might be able to locate the probe but won't get a good anatomical image around the probe. (The authors imply this when they mention an image artifact. Is there an image of it?) B) There is RF heating of the metal in the probe. C) There are magnetic forces on the probe (the authors even have a recent Science Advances paper on that subject).

3. The description of how the location is determined from the NMR measurements is not complete enough to be reproducible. There are no citations listed for the particle filter technique or how it can be used to determine the location of a dipole relative to a number of sensors. It is an interesting way to determine location, rather than trying to solve the inverse problem, and is a critical part of the paper. Has this been done before? Are there citations? What is the starting volume distribution for the 200 test particles in the Monte Carlo simulation? If this is iterative, how many iterations are used? What exactly do you mean by "resample the particles?"

4. I know that quantum sensors are all the rage right now, but a bulk NMR magnetometer is not a quantum sensor. There are no single spins, quantum entanglement, or quantum interference involved.

(Remarks to the Author)

This paper proposes a wireless magnetic tracking technology based on nuclear magnetic resonance, demonstrating significant advantages in terms of high sensitivity, miniaturization, and compatibility with MRI systems. The study validates the application potential of NMR magnetic sensors in different types of magnetic trackers (rigid, flexible, hollow, and ferrofluidic) through various experiments and highlights their broad applicability in minimally invasive medical devices. However, despite the novelty of this research, there are several aspects that require further discussion and refinement. Detailed review comments are as follows:

1. The current system utilizes a planar hexagonal sensor array, which simplifies the hardware design but imposes limitations on localization accuracy along the Z-axis. It is recommended that the authors further investigate the effect of sensor spacing in the Z-direction using a three-dimensional array configuration (e.g., Figure S11.G), in order to quantify the potential improvement in precision and support its clinical applicability.
2. Although individual NMR magnetic sensors are compact, the current sensor layout (as shown in figure S11) appears to limit the technology's application range, especially for instrument tracking outside the head region. The existing layout may cause interference between the sensors and the patient in other body areas, reducing clinical applicability. The authors are encouraged to explore optimized layout designs for different body regions to expand the technology's application to more clinical scenarios. Furthermore, as a general-purpose tracking system, requiring layout re-optimization for each body region may significantly complicate its clinical use. The authors are advised to design a highly adaptable universal layout to simplify clinical operations and improve the technology's potential for broader adoption.
3. The integration of magnetic trackers with MRI imaging is demonstrated in figure 3, showing that the magnetic tracking data aligns with the artifact of the guidewire in MRI images, enabling precise localization. However, the potential artifacts or interference caused by the magnetic tracker on MRI imaging are not thoroughly discussed. Additionally, there is a lack of comparative MRI images without magnetic trackers, making it difficult to assess their specific impact on imaging quality.
4. The dipole model currently assumes that the magnetic moment is aligned with the B_0 field (Equation 2). However, for flexible trackers, bending may cause deviations in the magnetic moment direction, which can introduce localization errors. It is encouraged to incorporate a corrected model that accounts for this effect to improve the accuracy of the particle filter-based tracking algorithm.
5. The paper mentions an ex vivo pig brain navigation experiment but does not provide accompanying video evidence. This omission may affect the reproducibility and credibility of the results. The authors are advised to provide video documentation of the experiment to visually demonstrate the navigation process and its correlation with MRI imaging, thereby enhancing the study's visualization and reliability.
6. In lines 59-76 and the supplementary material, the authors provide only a brief overview of the selection and construction methodology for the NMR sensors. There is insufficient explanation regarding the underlying principles of the localization method. It is recommended that the authors include a more detailed explanation of these fundamental principles. For instance, clarification on the Free Induction Decay (FID) signal and the rationale behind designing the NMR sensor with the RF coil encircling a glass tube of water would be beneficial. This addition will help readers grasp the principle-based innovations of this sensor system.
7. All experiments were conducted using a 7T preclinical animal MRI system (line 89 in the main text), whereas standard clinical MRI systems typically operate at 1.5–3T. Please clarify whether the proposed tracking method is compatible with clinical MRI field strengths.
8. The background magnetic field appears to have an effect on the localization results. If this is indeed the case, it is recommended that the authors conduct additional localization experiments with the NMR sensing system under varying background fields to investigate this influence.
9. While Figure S22 demonstrates dual-target localization, the experimental setup is limited to a relatively large inter-target distance (4 cm). It does not evaluate performance in scenarios involving close spacing or overlapping magnetic fields. It is suggested that the authors assess whether accurate dual-target tracking is achievable at smaller separations.
10. The authors dedicate significant space in the manuscript to describing various miniaturized tracker devices. At line 171 in the main text, an application scenario for the hollow magnetic tracker is mentioned. However, the authors primarily list the positioning performance of these different magnetic trackers within the NMR sensing system, without thoroughly discussing their practical application value. The authors are encouraged to supplement their study with experiments simulating real-world conditions (e.g., ex vivo or in vivo experiments, or integration with commercial equipment) to demonstrate the practical utility of these multiple forms of magnetic trackers.
11. The paper does not sufficiently demonstrate the advantages of NMR magnetic tracking over electromagnetic (EM) tracking. While EM tracking systems perform well in non-MRI environments, the strengths of NMR magnetic tracking lie in its compatibility with MRI and its high sensitivity. The authors are encouraged to conduct comparative experiments to quantify differences between the two technologies in terms of accuracy, sensitivity, miniaturization, and clinical applicability, thereby

better highlighting the unique value of NMR magnetic tracking.

12. With the continuous improvement of MRI imaging speeds, the relevance of this research warrants further discussion. Faster MRI imaging has alleviated some of the challenges associated with real-time navigation. The authors should further explore the potential synergy between NMR-based magnetic tracking and fast MRI imaging, emphasizing the unique advantages of magnetic tracking in terms of real-time performance and miniaturization.

13. The paper notes that this method is unsuitable for patients with ferromagnetic implants and faces challenges in environments with dynamic tissues (e.g., the heart or nerves) due to background signal variations. These limitations may hinder the adoption of this technology in complex clinical scenarios. The authors are encouraged to discuss potential solutions to address these limitations.

In summary, this paper introduces an innovative NMR-based wireless magnetic tracking technology with advantages in sensitivity, miniaturization, and MRI compatibility. While the research shows great potential, some areas need further improvement, such as optimizing sensor layouts, addressing MRI interference, refining models for flexible trackers, and providing more experimental evidence. Additionally, comparisons with electromagnetic tracking and discussions on clinical limitations and applications would enhance the study's impact and practical relevance.

Version 1:

Reviewer comments:

Reviewer #1

(Remarks to the Author)

I thank the authors for their additional data and thorough responses. I support publication of the manuscript in its current form.

Reviewer #2

(Remarks to the Author)

This paper has been well revised. I have no more comments.

Response Letter

We would like to thank the reviewers for their invaluable comments and suggestions on our manuscript. We have addressed the points raised by the reviewers point by point below, which significantly improved our paper. Our responses are in blue color, and all changes in the revised manuscript are highlighted with yellow color text.

Reviewer #1

Tiryaki et al. describe a novel way to perform magnetic tracking using a soft ferromagnetic marker and an array of NMR probes in an MRI instrument. I'll start by saying this is probably not the way I would go about tracking using an MRI, but it is an interesting way to do it and might have its advantages. The authors have done a solid job in their performance analysis and their consideration of many different effects, so I think it will be publishable in Nature Communications, but I do have some questions, concerns, and suggestions below that need to be addressed.

Response: We thank the reviewer for the kind comments.

Comment 1.1) I'm not completely clear about the use case for this device. Is this for identifying the location of a catheter during medical procedures, where one wants to overlay the location with an MRI image? Or is it more general and just using the MRI as a big magnet? One of the cited papers was tracking bees, for instance. If it is medical uses, there is a whole field of catheter tracking that has been ignored in the introduction. There are even commercially sold devices that do this for MRI with similar accuracy to the manuscript over a larger volume. I provide a short list here:

<https://ajronline.org/doi/10.2214/ajr.183.2.1830391>

<https://onlinelibrary.wiley.com/doi/10.1002/mrm.24654>

<https://onlinelibrary.wiley.com/doi/full/10.1002/mrm.20202>

<https://inventions.techventures.columbia.edu/technologies/wireless-magnetic--1438>

<https://www.robinmedical.com/endoscout.html>

I'd like the authors to better describe the advantages and weaknesses of their technique in relation to these, because their method is quite different, but the application is not unique.

Response: We thank the reviewer for their comment and the provided literature, which helped us further highlight the advantage of NMR magnetic tracking.

The primary use case and purpose of the proposed wireless magnetic tracking system is to provide rapid position feedback for minimally invasive devices, such as catheters, guidewires, and needles, within an MRI scanner. Consequently, we aim to develop an auxiliary wireless tracking system that operates with preoperative MR images in medical applications, and potentially with real-time MR images in the future. We envision that, in the future, MR technology will find increasing use as a therapeutic platform, driven by developments in imaging, MR-compatible robotic systems, and sensing methods, such as the proposed NMR magnetic tracking. In this sense, no, the MRI scanner is not only used as a big magnet.

Moreover, the current proposed NMR magnetic tracking system utilizes the RF hardware, in addition to the MRI scanner's main static field, to generate the NMR signal. Although we also acknowledge that we would undoubtedly benefit from the flexibility of use if we had separate RF hardware for the NMR sensors that synchronously operate with the MRI scanner.

All that being said, we also believe that NMR magnetic sensing, as a high-sensitivity magnetic field measurement technique, enabled us to highlight the true potential of wireless magnetic tracking, which was not previously possible to elucidate with the magnetic sensing methods used in magnetic tracking. Therefore, we also try to keep our use cases general, much like the Gleich *et al.*¹⁰ with their bee tracking example, while staying within reason, considering medical applications in MRI scanners. Furthermore, we also envision that NMR magnetic tracking could find future use outside of MRI scanners, with constructions such as the recently evolving single-sided magnetic resonance sensors²⁹. We also briefly mentioned this potential venue for NMR magnetic tracking in the "Discussion" section

Furthermore, while clinical use of high-field MRI scanners is becoming popular²⁶, the integration of NMR magnetic sensor arrays into lower field MRI scanners such as 1.5 T/3 T and emerging ultra-low-field MRI scanners could increase the affordability of magnetically guided minimally invasive operations in the future²⁷,

The Endoscout tracking system, as cited by the reviewer, performs localization by measuring magnetic field changes due to the magnetic gradient coils of the MRI scanner. It is composed of

three-axis 7.4 mm fluxmeter coils ¹⁵. In this sense, the Endoscout system can be classified as an example of an onboard magnetic sensing approach, which we introduced in the “Introduction”.

Among remote sensing approaches, commercial electromagnetic (EM) sensors ⁷ and onboard magnetic sensors ¹²⁻¹⁵ are the most common approaches.

However, this system also suffers from the same issues as other tethered onboard magnetic sensing approaches, such as larger dimensions, which prohibit integration in small medical tools.

Although the proposed magnetic tracking method is ultimately to complement, not replace, the MRI-based tracking methods. We agree with the reviewer that highlighting specific advantages and weaknesses enhances the value of our study. Therefore, as reviewers requested, we also evaluated the proposed NMR magnetic tracking method in Comparison to MR image-based tracking methods suggested by the reviewers, as well as others discussed in the “Discussion”.

Additionally, the NMR magnetic sensing technology addresses certain drawbacks of MR image-based tracking. Despite continuous improvements in MRI speed, real-time 3D tracking with passive fiducial markers, such as magnetic ²², 19-Fluorine ²³, and RF coils²⁴, and active MRI pickup coils ²⁵, remains limited by slice thickness ²² and RF-induced tissue heating. In contrast, NMR sensors provide continuous 3D tracking without requiring the deposition of RF energy (see Supplementary Information), enabling continuous navigation with no risk of heating. Furthermore, NMR magnetic sensing eliminates acoustic noise, which in fast MRI-based tracking can exceed safe exposure limits ^{3,5} and hinder communication between clinicians during interventional MRI operations. Because NMR does not rely on MR gradients (figure S23, Movie S8), it is inherently silent, creating a safer acoustic environment for both patients and clinicians. Moreover, NMR tracking also functions in air-filled cavities, where most MRI-based tracking methods, except ¹⁹F tracking ²³, fail due to the absence of ¹H of the surrounding water or tissue, allowing for the uninterrupted tracking of tools such as needles during insertion from outside the skin.

Comment 1.2) If the use case is to use the device in conjunction with MRI, there needs to be some discussion and analysis of how well the techniques can coexist as well as the safety implications. Ferromagnetic materials are usually considered incompatible with MRI for a number reasons. A) They distort the image, so they might be able to locate the probe but won't get a good anatomical image around the probe. (The authors imply this when they mention an image artifact. Is there an image of it?) B) There is RF heating of the metal in the probe. C) There are magnetic forces on the probe (the authors even have a recent Science Advances paper on that subject).

Response: We thank the reviewer for the comment. To address MRI compatibility and the safety implications, we have added the following content to the manuscript.

A) MRI artifact: The magnetic image artifact, which obstructs the anatomical image around the close vicinity of the magnetic trackers, is unavoidable for magnetic trackers in MR images. An example of image artifacts is presented in our previous studies [2022Tiryaki]; however, for the completeness of this work, we have added a video of guidewire navigation (Movie S9) and snapshots of the artifact (Figure S26) in the “Magnetic Artifact in MR Images” section of the Supplementary Materials.

S11. Magnetic Artifact in MR Images

Placing a magnetic tracker inside an MRI machine will disturb the magnetic field in the shape of a magnetic dipole in three-dimensional space. This disturbance is significant in a volume surrounding the tracker, leading to a substantial change in the NMR frequency of water, as stated in Eq. 1. As a result, the volume will not be excited within the utilized RF signal bandwidth, causing complete signal loss in that area. At greater distances from the tracker, the disturbance in the magnetic field is not strong enough to significantly shift the frequency. However, it can still distort the image by affecting the calculations of the gradient fields, which can show itself as a skewed image. Two ex vivo experiments using porcine brain were conducted to demonstrate this effect in a real tissue. In the first experiment, the tracker was inserted inside the brain in 10 steps, and an MR image was taken after each step. The entire process is illustrated in Figure S26 and summarized in Movie S9. In the second experiment, a preoperative image was first taken. Then, NMR sensors tracked the tracker's position throughout the insertion process. Finally, a postoperative image was taken. The preoperative and postoperative images are overlaid with the

estimated position of the tracker. They are shown in Figure S27 as snapshots of the process, and the whole procedure is documented in Movie S2.

B) RF Heating: The RF heating is a common problem for conductive objects during MRI, especially if the conductive object has a specific geometry, turning the object into an RF resonator. For instance, long conductive wires whose length is comparable to the wavelength of the MRI RF signal, or solenoid coils whose resonance frequency is tuned to the frequency of the MRI RF signal. The conductive magnetic trackers, particularly rigid iron trackers, that we used for NMR tracking are too small to exhibit resonance behavior at the RF frequencies used by MRI scanners. The soft elastic trackers and the ferrofluid ones, on the other hand, are not conductive enough to cause RF heating issues.

Moreover, we would like to emphasize that MRI-based RF heating issues arise during long-duration and continuous imaging. The NMR magnetic sensors do not radiate RF signals outside of their shielding; therefore, they do not cause any heating issues. To demonstrate the RF safety in terms of heating, we performed an RF heating experiment while MR imaging and NMR sensing in a saline medium with and without our trackers, and we reported our results in the supplementary information.

S13. RF heating experiments

To demonstrate that the NMR magnetic tracking does not cause RF-induced heating problems, we performed controlled imaging and sensing experiments. We compared the temperature increase during the extended imaging sequence and sensing in two different sets of experiments. First, the experiments were conducted using 80 ml of 2.6 M saline solution in imaging and sensing mode. A hollow tracker was then placed inside the solution, and the experiments were repeated. A Gradient Echo (GRE) sequence with a repetition time of 10 ms, an RF pulse duration of 0.54 ms, and a power of 90 W was used for imaging mode, while the sensing mode remained unchanged. Each experiment was conducted over 20 minutes, and sufficient time was allowed for the system to stabilize between each experiment. Three temperature sensors were utilised during each experiment, two placed next to the tracker and one far away to measure the environment's temperature. During the imaging period, we observed a consistent temperature increase due to the high conductivity of the solution (see Figure S29). However, we noted no significant change in

temperature after placing the hollow tracker inside the solution. This is expected because the geometry of the tracker is much smaller than the signal wavelength, resulting in a practically negligible absorption rate. Additionally, no substantial temperature change was detected in the sensor mode, which reflects our expectation that no RF radiation exists outside the sensor shielded box.

Figure S29. Rf heating comparison in imaging and sensing mode inside MRI. Temperature measurement of 80 ml of 2.6 M saline solution over 20 minutes (A) and with a hollow tracker inside the solution (B) in imaging and sensing modes.

We also include additional discussion about RF heating in the “Discussion” section, which further highlights the advantage of the proposed NMR magnetic tracking method.

Additionally, the NMR magnetic sensing technology addresses certain drawbacks of MR image-based tracking. Despite continuous improvements in MRI speed, real-time 3D tracking with passive fiducial markers, such as magnetic ²², 19-Fluorine ²³, and RF coils²⁴, and active MRI pickup coils ²⁵, remains limited by slice thickness ²² and RF-induced tissue heating. In contrast, NMR sensors provide continuous 3D tracking without requiring the deposition of RF energy (see Supplementary Information), enabling continuous navigation with no risk of heating.

C) **Magnetic force and torques:** We added a “Magnetic Force and Torque Safety” section in the Supplementary Information.

S14. Magnetic force and torque safety

The magnetic force and torque are, in general, important safety risks in MRI-compatible devices. We observe magnetic forces at the entrance of the MRI bore due to the high magnetic gradients, which could reach 20 T/m in 7 Tesla MRI scanners, and magnetic torques at the imaging center of the MRI scanner, where we have high uniform magnetic fields. We calculated the maximum magnetic force exerted on the tracker with a 1 emu magnetic moment using

$$F = \mu_0 V M_s \frac{\partial H_z}{\partial z}, \quad [S20]$$

as 19 mN in our 7 T MRI scanner⁶. A human operator or a mechanical insertion mechanism could easily overcome this force. Moreover, the magnetic force on the magnetic tracker disappears once the tracker is at the imaging center. The MR imaging gradient would apply a small force on the tracker; however, due to the very low strength of these gradients, typically less than 60 mT/m, the magnetic force during imaging would be in the order of or less than a couple of μN , which would barely vibrate the tip of a guidewire. Since our NMR magnetic tracking does not utilize gradient coils, we do not need to consider imaging gradient-based forces either.

On the other hand, the magnetic field at the center of the MRI scanner applies a magnetic torque, aligning the tracker’s magnetic easy axis with the field direction. Since we used soft magnetic materials rather than hard magnetic materials, this magnetic torque is much smaller than in our previous work. We calculated the maximum torque acting on the 1 emu tracker with a 0.6 mm diameter and a 2 mm height solid tracker, using⁶

$$\tau = \frac{\mu_0 V N M_s^2}{2}. \quad [S21]$$

For the aspect ratio of 3.33, we calculated N from Figure S4c. Then the maximum torque can be calculated as 0.3 mN/m, which is an order of magnitude smaller than the torques in our previous work. A flexible tracker with the same 1 emu has much lower torque since the magnetic materials are distributed over a larger volume with lower magnetic density. That being said, we should note

that there is no literature on the calculation of magnetic torque on flexible magnetic materials composed of magnetic microparticles and an elastomer matrix at under-saturation conditions.

Comment 1.3) The description of how the location is determined from the NMR measurements is not complete enough to be reproducible. There are no citations listed for the particle filter technique or how it can be used to determine the location of a dipole relative to a number of sensors. It is an interesting way to determine location, rather than trying to solve the inverse problem, and is a critical part of the paper. Has this been done before? Are there citations? What is the starting volume distribution for the 200 test particles in the Monte Carlo simulation? If this is iterative, how many iterations are used? What exactly do you mean by “resample the particles?”

Response: We thank the reviewer for the comment. To increase the reproducibility of the proposed tracking method, we have elaborated the description of the particle filter in the “Materials and Methods” section in the supplementary materials by adding citations to particle filter techniques. To the best of our knowledge, it is the first time a particle filter is used for magnetic dipole tracking; however, particle filtering is used for dipole-based tracking in array-based electromagnetic tracking³³. We also provided more details on our implementation of the particle filter.

A particle filter is a recursive Bayesian state estimation technique that uses the Monte Carlo method to approximate the posterior distribution of the estimated state in three steps: propagation, reweighting, and resampling³². It has been used in the localization of dipole-based EM trackers³³. The algorithm first approximates the state distribution using N particles, which are distributed uniformly over the state space. Then, in the propagation step, it predicts the next time evolution of each particle using a prediction model with noise. Next, in the reweighting step, it approximates the posterior distribution of the state by calculating the probability of each particle contributing to the observed sensor measurement using a measurement model with noise. Finally, in the resampling step, the algorithm samples N new particles from the approximated posterior distribution and replaces the previous set of N particles with new particles. Then, the algorithm recursively repeats itself for each time step with new sensor data. This method recursively refines the estimated position using the obtained sensor data, generally converging within 1-2 time steps if N is sufficiently large. To ensure convergence in the first step, a large initial particle number, N_0 , is used. Then, in the next step, the particle number is reduced to N during the resampling step. To estimate the magnetic tracker position, we employed a particle filter-based state estimation model with the state, $\mathbf{x} = [\mathbf{r}^T, \Delta B] \in \mathbb{R}^4$, composed of tracker position and monitored background

magnetic field variation. We consider the tracking problem as a static prediction model and a nonlinear measurement model:

$$\mathbf{x}(k) = \mathbf{x}(k-1) + \mathbf{v}(k-1), \quad [\text{S8}]$$

$$\mathbf{z}(k) = \mathbf{B}_d(k) + \Delta B(k) + \omega(k), \quad [\text{S9}]$$

where k is the time step of estimation, \mathbf{z} is the measurement vector $\in \mathbb{R}^8$, and \mathbf{v} and ω are the prediction and measurement noise modeled as a Gaussian distribution with covariance $\Sigma_v = \text{diag}([\sigma_x, \sigma_y, \sigma_z, \sigma_{\Delta B}]) \in \mathbb{R}^{4 \times 4}$ and $\Sigma_\omega = \text{diag}([\sigma_{s0}, \sigma_{s1}, \dots, \sigma_{s7}]) \in \mathbb{R}^{8 \times 8}$. We used the prediction noise with $\sigma_x = \sigma_y = \sigma_z = 1$ mm and $\sigma_{\Delta B} = 0.1$ nT. The measurement noise is calculated during the sensor array calibration. $\mathbf{B}_d(k) = [B_d(\mathbf{r}_0), B_d(\mathbf{r}_1), \dots, B_d(\mathbf{r}_7)]^T \in \mathbb{R}^8$ is the vector of the sensor measurement model of the sensors at time step k . Note that the reference sensor is included in the estimation as the 0th sensor to estimate the change in the background field.

We started the particle filter with $N_0 = 20000$, and reduced the particle number to $N = 500$ after the first iteration of estimation. First, the algorithm propagates the particle position and background predictions using Eqn S6. Next, we calculate the expected sensor measurement, $\bar{\mathbf{z}}$, for each particle using Eqn S7 with the noise term. Then, we calculate the measurement likelihood for each particle using a Normal distribution,

$$P_i(\bar{\mathbf{z}}_i | \mathbf{z}) = N(\bar{\mathbf{z}}_i | \mathbf{z}, \Sigma_\omega), \quad [\text{S10}]$$

where i is the index of the particle in N particles. Later, we scale the particle probabilities, $\beta_i = \frac{P_i}{\sum_0^N P_i}$, between 0 and 1. Finally, we resampled the particles using a random number r from a uniform distribution (0,1). To avoid prediction errors due to the inhomogeneity effect in proximity to the sensors, if any sensor measurement exceeds 20 μT , we remove the measurement of the corresponding sensor, exclude it from the resampling stage of the particle filter, and estimate the position using the remaining sensors.

Comment 1.4) I know that quantum sensors are all the rage right now, but a bulk NMR magnetometer is not a quantum sensor. There are no single spins, quantum entanglement, or quantum interference involved.

Response: We appreciate the reviewer's point. NMR magnetometers exploit the quantum property of nuclear spin precession in a magnetic field. The Larmor precession, the quantization of spin energy levels, and coherent manipulation via RF pulses are fundamentally quantum mechanical in nature. Therefore, we refer to it as a quantum sensor. We updated our description of the quantum sensor in the manuscript to clarify what we refer to as a quantum sensor as follows.

The required sensitivity could be achieved with quantum sensing approaches exploiting quantum properties of matter for magnetic field measurement ¹⁶, such as superconducting quantum devices (SQUID) ¹⁷, optically pumped magnetic (OPM) sensors ¹⁸, electron spin magnetic resonance ¹⁹, and nuclear magnetic resonance (NMR) magnetic sensors ²⁰.

After carefully investigating the literature on magnetic sensing methods, we observed that only sensing methods based on quantum properties, such as SQUID, OPM, ESMR, and NMR sensors, have the potential to provide the required sensitivity for magnetic tracking on smaller scales. We want to use our work as an opportunity to introduce the magnetic tracking problem to the quantum sensor community.

Reviewer #2

This paper proposes a wireless magnetic tracking technology based on nuclear magnetic resonance, demonstrating significant advantages in terms of high sensitivity, miniaturization, and compatibility with MRI systems. The study validates the application potential of NMR magnetic sensors in different types of magnetic trackers (rigid, flexible, hollow, and ferrofluidic) through various experiments and highlights their broad applicability in minimally invasive medical devices. However, despite the novelty of this research, there are several aspects that require further discussion and refinement. Detailed review comments are as follows:

Response: We thank the reviewer for the kind comments.

Comment 2.1) The current system utilizes a planar hexagonal sensor array, which simplifies the hardware design but imposes limitations on localization accuracy along the Z-axis. It is recommended that the authors further investigate the effect of sensor spacing in the Z-direction using a three-dimensional array configuration (e.g., Figure S11.G), in order to quantify the potential improvement in precision and support its clinical applicability.

Response: We appreciate the reviewer's comment. As suggested, we have expanded the "S5. Workspace Scaling" section to include various 3D configurations and have analyzed their corresponding precision maps, which are shown in the new Figure S.12. Additionally, we prepared "Movie S10" to enhance the visualization of the 3D precision map of the arrays. We used the experimental floor noise to estimate the precision map for all 3D configurations in three-dimensional space. The following part was added to "S5. Workspace Scaling":

We can achieve a significantly larger workspace by distributing additional sensors in 3D space. In our study, we illustrated various 3D sensor array configurations in different medical intervention scenarios, focusing on three specific regions of the body while utilizing the experimental noise floor and the mentioned simulation techniques. First, we examined endonasal routes to the skull base using an array of 19 sensors arranged in a triangular format across three rings to cover the average human head. One ring is positioned at the top of the head, while the other two rings encircle the forehead and nasal areas. As shown in Figure S12a-c, the 2D precision map indicates that most brain areas have a precision of more than half a millimetre, and there is over 1 mm of

precision from the nasal area to the brain at a depth of up to 3 cm. As illustrated in Movie S10, the brain region maintains the half-millimeter precision throughout the entire brain volume.

Next, we analyzed transfemoral access via the common femoral artery, located about 2 cm below the inguinal ligament. For this analysis, we used the same six-sensor ring array around the thigh, demonstrating the extensive workspace available (Figure S12d-f). The precision map indicates that we can achieve better than 1 mm accuracy up to 7 cm deep into the tissue until we reach the bone. By cascading the rings, we can cover a much larger volume if necessary; in this case, we utilized a total of 24 sensors arranged in four arrays.

Finally, we investigated the abdominal area, the largest region. Numerous procedures may require access to internal structures, such as suprapubic cystostomy, which targets the anterior bladder wall; peritoneal dialysis (PD) catheter insertion into the peritoneal cavity; or percutaneous nephrostomy, which provides access to the kidneys from the posterior flank. To ensure access to the entire abdominal volume, we repeated the triangular configuration three times in succession, creating an array of 39 sensors, which formed a belt around one side of the abdomen (either posterior or anterior access). As evident in the 2D precision slice taken 7 cm deep in the body (Figure S12g-i), we achieved a precision of over 1 mm across the entire 40 cm width of an average body. By cascading this pattern, we can effectively cover the whole abdominal area, rather than focusing on just one specific procedure. All the analyzed areas are represented in 3D space, with the results showcased in Movie S10.

Figure S11. Workspace scaling. A) Current sensor array arrangement. B-C) Estimation precision as a function of tracker distance (B) and sensor spacing (C). D) Sensor number scaling. E-F) Comparison of estimation precision with different sensor numbers as a function of distance from the tracker.

Figure S12. Various 3D sensor arrays for medical interventions. A-C) A 3D array proposed for the head, with its precision map separate (B) and overlaid (C) on the head. D-F) A 3D array proposed for the thigh region and its precision map, separate (E) and overlaid (F) on the thigh. G-I) A 3D array proposed for the abdominal region and its precision map, separate (H) and overlaid (I) on the abdominal region.

Comment 2.2) Although individual NMR magnetic sensors are compact, the current sensor layout (as shown in Figure S11) appears to limit the technology’s application range, especially for instrument tracking outside the head region. The existing layout may cause interference between the sensors and the patient in other body areas, reducing clinical applicability. The authors are encouraged to explore optimized layout designs for different body regions to expand the technology’s application to more clinical scenarios. Furthermore, as a general-purpose tracking system, requiring layout re-optimization for each body region may significantly complicate its clinical use.

Response: We thank the reviewer for the comments. We realized that we did not mention the aluminum RF shielding we applied around our sensors, which is, in fact, very important for the reproducibility of our results. The aluminum shielding around the sensors isolates the RF signal of each NMR coil, preventing interference between sensors and with the tissue. We introduced the RF shield in the “Result” part with the following.

Each sensor is placed in an RF shielding composed of aluminium foil (figure S15a), which eliminates interference between sensors and also the nearby tissue (see Supplementary Information).

Moreover, to demonstrate that the sensors do not receive any NMR signal from adjacent sensors or nearby tissues due to the shielding, we perform MR imaging experiments with two sensors placed close to a brain tissue. We acquire two sets of slice images: one from a plane passing through the sensors, and one from a plane passing through the brain, as shown in Figure S28a. We observed that due to the shielding on one of the sensors, the MR images captured from each RF coil did not receive any signal from the water of the other RF coil (Figure S28b-c). From the second set of slice images, we observed that while the shielded sensor did not receive any signal from the brain tissue, leaving only imaging noise (Figure S28d), the unshielded sensor received a significant signal from the tissue (Figure S28e). This difference demonstrated that our RF shield effectively eliminates interference with the patient. We have added these experiments to the Supplementary Information, in a new section titled “RF Shielding”.

S12.RF Shielding

Each sensor consists of a miniaturised RF coil, which allows it to excite the volume inside and around the coil. This can lead to unintended excitation of nearby sensors or tissues. To minimize potential interference from the surrounding environment, the sensor is placed inside an aluminium box serving as an RF shield. To demonstrate the efficacy of the RF shield, we placed two NMR sensors, a shielded and an unshielded sensor, on top of a porcine brain, as shown in Figure S28a. We performed 2D MR imaging experiments using the Localizer sequence with an excitation power of 80 microwatts, which matches the power used in NMR magnetic sensing. We acquire two sets of slice images: one from a plane passing through the sensors, and one from a plane passing through the brain, as shown in Figure S28a. We observed that due to the shielding on one of the sensors, the MR images captured from each RF coil did not receive any signal from the water of the other RF coil (Figure S28b-c). From the second set of slice images, we observed that while the shielded sensor did not receive any signal from the brain tissue, leaving only imaging noise (Figure S28d), the unshielded sensor received a significant signal from the tissue (Figure S28e). This experiment confirms the shield's effective role and shows no interference between sensors while shielded.

As reviewers' suggestion, we also explored different sensor array layouts, including tracking in the abdominal area, Head region, and thigh, which is added to the "S5. Workspace Scaling" in the supplementary material. Please see Response to Comment 2.1.

Figure S28. NMR sensor interference with the environment. A) The setup used to study the extent of the captured NMR signal with and without shielding. B-C) Sensor head image in the upper coronal slice (slice 1) with the shielded sensor (B) and the unshielded sensor. D-E) Image of the lower coronal slice (slice2) with the shielded sensor (D) and unshielded sensor (E).

Comment 2.3) The authors are advised to design a highly adaptable universal layout to simplify clinical operations and improve the technology's potential for broader adoption.

Response: We agree with the reviewer on the potential benefit of an adaptable universal sensor layout for future applications; however, we believe such a design requires a separate focus study. Therefore, we highlighted the potential advantage of such an adaptable universal sensor layout in our “Discussion” section as follows.

It is possible to create denser NMR magnetic arrays with more sensors or distribute the sensor arrays with modular layouts around a desired volume to increase the workspace without interference from one another.

Comment 2.4). The integration of magnetic trackers with MRI imaging is demonstrated in figure 3, showing that the magnetic tracking data aligns with the artifact of the guidewire in MRI images, enabling precise localization. However, the potential artifacts or interference caused by the magnetic tracker on MRI imaging are not thoroughly discussed. Additionally, there is a lack of comparative MRI images without magnetic trackers, making it difficult to assess their specific impact on imaging quality.

Response: We thank the reviewer for the valuable suggestion. To give the audience a general idea of what a magnetic tracker artifact would look like in MR images, we repeated the tracking experiment in porcine brain tissue first with MR imaging and second with NMR sensing. We added the result in the “S11. Magnetic Artifact in MR Images”. We created videos during both experiments (Movies S2 and S9). We also presented the magnetic artifacts in MR image snapshots in Figure S26 from Video S9.

S11. Magnetic Artifact in MR Images

Placing a magnetic tracker inside an MRI machine will disturb the magnetic field in the shape of a magnetic dipole in three-dimensional space. This disturbance is significant in a volume surrounding the tracker, leading to a substantial change in the NMR frequency of water, as stated in Eq. 1. As a result, the volume will not be excited within the utilized RF signal bandwidth, causing complete signal loss in that area. At greater distances from the tracker, the disturbance in the magnetic field is not strong enough to significantly shift the frequency. However, it can still

distort the image by affecting the calculations of the gradient fields, which can show itself as a skewed image. Two ex vivo experiments using porcine brain were conducted to demonstrate this effect in a real tissue. In the first experiment, the tracker was inserted inside the brain in 10 steps, and an MR image was taken after each step. The entire process is illustrated in Figure S26 and summarized in Movie S9. In the second experiment, a preoperative image was first taken. Then, NMR sensors tracked the tracker's position throughout the insertion process. Finally, a postoperative image was taken. The preoperative and postoperative images are overlaid with the estimated position of the tracker. They are shown in Figure S27 as snapshots of the process, and the whole procedure is documented in Movie S2.

Figure S26. Tracker MR images during insertion inside a porcine brain. Tracker artifacts during the insertion inside a porcine brain in oblique and sagittal planes.

Comment 2.5) The dipole model currently assumes that the magnetic moment is aligned with the B_0 field (Equation 2). However, for flexible trackers, bending may cause deviations in the magnetic moment direction, which can introduce localization errors. It is encouraged to incorporate a corrected model that accounts for this effect to improve the accuracy of the particle filter-based tracking algorithm.

Response: We thank the reviewer for the comment. We agree that a more elaborate magnetic model, such as a discretized dipole model, could improve the tracking accuracy of the flexible tracker. However, we believe that such a complex model requires a more focused investigation in a separate study, which might be more suitable for a technical journal. Therefore, we preferred to leave it for future work, while providing a benchmark for tracking accuracy based on the basic dipole assumption.

Comment 2.6) The paper mentions an ex vivo pig brain navigation experiment but does not provide accompanying video evidence. This omission may affect the reproducibility and credibility of the results. The authors are advised to provide video documentation of the experiment to visually demonstrate the navigation process and its correlation with MRI imaging, thereby enhancing the study's visualization and reliability.

Response: We appreciate the reviewer's valuable suggestion, which enhanced our comparative study. To clearly illustrate the process of positioning the magnetic tracker in the ex vivo experiment and compare it with the tracker artefact, a new section, "S11. Magnetic Artefact in MR Images", was added to the supplementary material in which two sets of experiments, one with MR images and one with NMR sensing, are demonstrated. In the tracking experiment, a preoperative image was first captured, and after tracking throughout the operation with NMR sensors, a postoperative image was taken. This process is documented in Movie S2, with snapshots presented in Figure S27 to demonstrate the navigation process visually.

S11. Magnetic Artifact in MR Images

Placing a magnetic tracker inside an MRI machine will disturb the magnetic field in the shape of a magnetic dipole in three-dimensional space. This disturbance is significant in a volume surrounding the tracker, leading to a substantial change in the NMR frequency of water, as stated in Eq. 1. As a result, the volume will not be excited within the utilized RF signal bandwidth, causing complete signal loss in that area. At greater distances from the tracker, the disturbance in the magnetic field is not strong enough to significantly shift the frequency. However, it can still distort the image by affecting the calculations of the gradient fields, which can show itself as a skewed image. Two ex vivo experiments using porcine brain were conducted to demonstrate this effect in a real tissue. In the first experiment, the tracker was inserted inside the brain in 10 steps, and an MR image was taken after each step. The entire process is illustrated in Figure S26 and summarized in Movie S9. In the second experiment, a preoperative image was first taken. Then, NMR sensors tracked the tracker's position throughout the insertion process. Finally, a postoperative image was taken. The preoperative and postoperative images are overlaid with the estimated position of the tracker. They are shown in Figure S27 as snapshots of the process, and the whole procedure is documented in Movie S2.

Figure S27. Tracker navigation inside a porcine brain with pre- and postoperative images.

Comment 2.7) In lines 59-76 and the supplementary material, the authors provide only a brief overview of the selection and construction methodology for the NMR sensors. There is insufficient explanation regarding the underlying principles of the localization method. It is recommended that the authors include a more detailed explanation of these fundamental principles. For instance, clarification on the Free Induction Decay (FID) signal and the rationale behind designing the NMR sensor with the RF coil encircling a glass tube of water would be beneficial. This addition will help readers grasp the principle-based innovations of this sensor system.

Response: We thank the reviewer for their valuable suggestion. We added an extra section, “NMR magnetic sensing,” in the Supplementary Information to further explain the fundamentals of NMR signal generation and provide principles on why the sensor has the mentioned structure.

NMR magnetic sensing The basis of NMR magnetic sensing resides in the precession of nuclear spins in the presence of a background magnetic field (B_0) after excitation by an oscillating external magnetic field. In the presence of B_0 , the net magnetisation of all the spins will redirect in parallel with the field. By superimposing an oscillating magnetic field perpendicular to the background field (called B_1), the spins start to precess around the B_0 axis with a frequency close to the Larmor frequency, which depends on the gyromagnetic ratio of the specific nuclei and the background magnetic field :

$$\gamma = \frac{f}{B_0 (T)} \quad [S1]$$

where γ , f and B_0 are the gyromagnetic ratio (Hz/T), the Larmor frequency (Hz), and the background magnetic field (T), respectively.

This precession exhibits a decreasing amplitude of latitude and is characterized as a decaying oscillating signal. This decaying oscillating signal read by an RF coil is called a free induction decay (FID) signal³¹. As stated in Eq. 1, the resulting frequency is dependent on the magnitude of the background field (B_0), so any change in the B_0 will affect the frequency of the FID signal, which later can be translated to the change in magnetic field:

$$f = \gamma B \rightarrow (f_0 + \delta f) = \gamma(B_0 + \delta B) \rightarrow \delta B = \gamma^{-1} \delta f \text{ where } \delta f = \frac{\Delta\phi}{\Delta t} \quad [S2]$$

Deionized (DI) water enclosed in a glass capillary was selected as the base structure for generating NMR signals, preventing evaporation over long periods and serving as a solid base for RF coil

fabrication. This choice ensures that the free induction decay (FID) frequency is compatible with the RF equipment in commercial scanners, allowing for consistent and repeatable results. To minimise the nonlinear magnetic effects of a dipole field across large volumes, the amount of water used was kept as small as possible. At the same time, ensuring the scanner's analogue-to-digital converter (ADC) could detect the FID signal's amplitude within a reasonable acquisition time. Finally, an RF transceiver composed of a multi-turn pickup coil and a tuning and matching circuit was constructed around the glass tube to excite the spins and detect the NMR signal in the form of FID.

Comment 2.8) All experiments were conducted using a 7T preclinical animal MRI system (line 89 in the main text), whereas standard clinical MRI systems typically operate at 1.5–3T. Please clarify whether the proposed tracking method is compatible with clinical MRI field strengths.

Response: We thank the reviewer for the comment. Although we have demonstrated NMR magnetic tracking in a 7T MRI scanner, the same physical principles would be valid for lower

field strengths, such as 3T and 1.5T, with a corresponding reduction in signal-to-noise ratio. We have added a related discussion on the reduction in signal level and its effect on tracking performance in different fields to the supplementary material.

Furthermore, while clinical use of high-field MRI scanners is becoming popular²⁶, the integration of NMR magnetic sensor arrays into lower field MRI scanners, such as 1.5 T and 3 T, and emerging ultra-low-field MRI scanners could increase the affordability of magnetically guided minimally invasive operations in the future²⁷; However, we must note that the magnetic sensing precision will decrease with the static field strength since signal-to-noise ratio in MRI scanners scale with $\propto B_0^{7/4}$ ²⁸. Moreover, NMR magnetic sensors constructed in a single-sided MR configuration could allow the use of magnetic tracking outside of the MRI scanners²⁹.

Comment 2.9) The background magnetic field appears to have an effect on the localization results. If this is indeed the case, it is recommended that the authors conduct additional localization experiments with the NMR sensing system under varying background fields to investigate this influence.

Response: Yes, the reviewer is correct. The variation of the background field does affect the tracking accuracy. Therefore, we placed the eight NMR magnetic sensors as a reference sensor far

behind the sensor array, as shown in Figure S9. As the distance to the magnetic tracker increases substantially, this sensor enables us to monitor variations in the background field. We use the variation over time in our position tracking algorithm. To clarify this, we add the following in the “Result” section.

We also introduced a reference NMR magnetic sensor placed far from the other sensors to monitor the variation of the background field, which could change substantially over a long duration of operation (figure S9).

Additionally, in the supplementary information, we clarified that background field variations have been monitored.

To estimate the magnetic marker position, we used a particle filter-based state estimation model with the state, $\mathbf{x} = [\mathbf{r}^T, \Delta B] \in \mathbb{R}^4$, composed of tracker position and monitored background magnetic field variation.

Comment 2.10) While Figure S22 demonstrates dual-target localization, the experimental setup is limited to a relatively large inter-target distance (4 cm). It does not evaluate performance in scenarios involving close spacing or overlapping magnetic fields. It is suggested that the authors assess whether accurate dual-target tracking is achievable at smaller separations.

Response: We appreciate the reviewer's comment. To further evaluate the performance of our two-tracker system, we conducted three additional experiments under the same setup, varying the inter-target distance from 4 cm to 2 cm and then to 1 cm. We added the results in the Supplementary Information.

We conducted tracking experiments using two trackers to explore the possibility of a multi-tracker system with the current number of sensors. A stage featuring parallel channels (Figure S10a) was used to move the trackers within the workspace (Figure S24 and S25). The experiment was conducted at three different distances: 4cm, 2cm, and 1cm. In each experiment, one tracker was moved 1 cm using a Teflon guidewire, while the other remained stationary. To estimate the positions using the particle filter, we increased the number of estimated states in the algorithm from three to six to account for the second dipole. The particle filter can estimate the positions of multiple trackers simultaneously; however, the accuracy and precision of these estimates decrease compared to single-dipole experiments. When the trackers are spaced 20 mm and 40 mm apart, the particle filter can still estimate their positions within a workspace of 60 mm and 40 mm away from the sensor array (Figure S24). In contrast, for single tracker scenarios, the workspace, with accuracy exceeding 2 mm, extends to 80 mm; however, as trackers are placed closer to each other, at 10 mm, the dipole field of two trackers becomes indistinguishable for the particle filter, leading to a further reduction in position estimation accuracy (Figure S25). We believe that this problem could be addressed in future robotics systems by incorporating system dynamics into the prediction stage of a particle filter.

Figure S24. Multi-bead experiment with a large distance between trackers. The position estimation accuracy of two trackers at a large proximity to each other, in 4 cm (A) and 2 cm (B), is investigated.

Figure S25. Multi-bead experiment with a small distance between trackers. The position estimation accuracy of two trackers at a distance of 1 cm in the left (A) and right (B) planes is investigated.

Comment 2.11) The authors dedicate significant space in the manuscript to describing various miniaturized tracker devices. At line 171 in the main text, an application scenario for the hollow magnetic tracker is mentioned. However, the authors primarily list the positioning performance of these different magnetic trackers within the NMR sensing system, without thoroughly discussing their practical application value. The authors are encouraged to supplement their study with experiments simulating real-world conditions (e.g., ex vivo or in vivo experiments, or integration with commercial equipment) to demonstrate the practical utility of these multiple forms of magnetic trackers.

Response: We thank the reviewer for the comment. We agree with the reviewer that in vivo experiments simulating the clinical condition would add value to our study. Unfortunately, at the current stage, we lack the experimental capabilities to demonstrate in vivo results. Therefore, we focused on ex vivo experiments. The magnetic tracking experiments in an ex vivo porcine brain aim to simulate the operation conditions of an MRI-guided needle insertion operation.

Later, we illustrated wireless tracking in an *ex vivo* porcine brain, simulating a needle insertion operation into the brain (Figure 3b). We inserted a straight catheter in the brain and took pre-operational MR images of the brain using an imaging surface coil shown in Figure 3c-d. Next, we manually switched the RF connection from the surface coil to the NMR sensor array. Then, we calibrated the background field with the diamagnetic brain tissue, which caused 1-2 μT variation on different sensors (Figure S15), and tracked the guidewire in the catheter in real time with NMR magnetic tracking (Movie S2).

To further demonstrate the potential of commercial medical equipment, as the reviewer suggested, we prepared a custom-made endoscope camera with a magnetic tracker, and we demonstrated tracking in an ex vivo porcine esophagus while performing endoscopic imaging.

Next, we integrated the miniaturized tracker with 1 emu magnetic moment into a custom-made endoscope camera system, and performed tracking in *ex vivo* on a porcine oesophagus to demonstrate wireless tracking while performing endoscope video capturing (see Movie S4 and Figure S17).

Figure S16. Endoscope camera tip tracking inside the porcine oesophagus. A) Custom made endoscope camera system and the Porcine oesophagus used for the operation. B) endoscope camera feed inside the oesophagus. C) Axial view of position tracking throughout the whole operation. D) Coronal view of position tracking throughout the whole operation.

Comment 2.12) The paper does not sufficiently demonstrate the advantages of NMR magnetic tracking over electromagnetic (EM) tracking. While EM tracking systems perform well in non-MRI environments, the strengths of NMR magnetic tracking lie in its compatibility with MRI and its high sensitivity. The authors are encouraged to conduct comparative experiments to quantify differences between the two technologies in terms of accuracy, sensitivity, miniaturization, and clinical applicability, thereby better highlighting the unique value of NMR magnetic tracking.

Response: We thank the reviewer for the comment. While Magnetic and EM tracking both use dipole-based models for tracking, they are distinct in nature due to the magnetic moment generation method by the tracker. Magnetic tracking uses passive magnetic trackers, while EM tracking needs to induce a current in the tracker. Therefore, a direct experimental comparison highlighting the advantage of magnetic tracking over EM and vice versa, similar to one presented by Gleich et al.¹⁰, between mechanical and electrical EM trackers, would be controversial for magnetic and EM trackers. Moreover, as the reviewer stated, these sensors address the tracking problem in two different operating conditions: low (Non-MRI) and high (MRI) background magnetic fields, both of which have strong clinical potential.

That being said, we agree with the reviewer that a meta comparison that places the NMR magnetic tracking method next to previous methods in terms of accuracy and miniaturization would be beneficial for benchmarking. Therefore, we have already provided a comparison plot (Figure S13) for miniaturization with a function of workspace and a Table for accuracies (Table 1) in SI, which includes EM tracker examples from the literature. We have updated the Table to refer to the EM trackers more clearly and changed the color of EM tracker dots in Figure S13.

Table 1. Comparison of wireless tracking systems *Statistics have not been reported explicitly. MT: magnetic tracking, MGT: magnetic gradient tracking, EMT: electromagnetic tracking.

	Tracking Modality	Tracker Size (mm ²)	Temp. Res.	Penet. Depth	Tracking Acc.
Son ¹¹	MT	12.8 × 6.4Ø	5 ms	5 cm	2.1 mm
Gleich ¹⁰	EMT	1.9 × 0.8Ø	25 ms	30 cm	2 mm at 10 cm*
Sharma ¹²	MGT	20 × 8Ø	300 ms	40 cm	2 mm at 12 cm*
Osaki ⁹	EMT	15 × 4Ø	100 ms	15 cm	<1 mm at 15 cm
Arx ¹³	MGT	6 × 3Ø	100 ms	10 cm	<1 mm at 10 cm
This work	MT	2 × 0.6 Ø	30 ms	8 cm	2 mm at 8 cm

Figure S13 Miniaturization comparison. It can be seen that in this work, we can track smaller volumes of magnetic material with better accuracy in a similar workspace ⁹⁻¹³. The lower limit of tracking accuracy is considered to be 2mm. Blue dots represent magnetic tracking methods, while green dots represent EM tracking methods.

To highlight the advantages of NMR magnetic tracking over wireless EM tracking, we have added the following section to the Discussion section.

Besides, NMR magnetic sensing provides a larger workspace compared to previous magnetic tracking methods¹¹ and a comparable workspace to wireless EM trackers of the same size¹⁰, with similar tracking accuracy (Table 1, Figure S13), without the RF-induced heating risks. Although our demonstrations have been limited by the size of our preclinical MRI scanner, the NMR magnetic sensing technology offers a potentially larger workspace due to the relatively small size of NMR magnetic sensors compared with planar coils used in EM trackers¹⁰. It is possible to create denser NMR magnetic arrays with more sensors or distribute the sensors around a desired volume to increase the workspace without interference from one another. Increasing sensor numbers could also be used to increase tracking accuracy further; however, this will limit the temporal resolution of tracking and require parallel hardware instead of the current serial hardware. Importantly, NMR magnetic sensing also addresses the position-calibration challenges that EM tracking systems face during integration with medical imaging. The gradient-based sensor positioning enables spatial calibration without the need for external hardware, allowing the magnetic tracker to be registered directly onto MR images (Figure 2f) and enabling single-stage, real-time positioning of stereotactic neurosurgery robots⁴. Finally, NMR trackers do not suffer from the dead-angle issues inherent to EM trackers, since NMR tracking does not require external excitation⁹.

Comment 2.13) With the continuous improvement of MRI imaging speeds, the relevance of this research warrants further discussion. Faster MRI imaging has alleviated some of the challenges associated with real-time navigation. The authors should further explore the potential synergy between NMR-based magnetic tracking and fast MRI imaging, emphasizing the unique advantages of magnetic tracking in terms of real-time performance and miniaturization.

Response: We thank the reviewer for the comment. We agree that the improvement in MRI speed will alleviate some real-time navigation challenges, but not address all the drawbacks of MRI-based tracking.

Firstly, 3D tracking will remain a challenge with MR image-based tracking, since the tracked object may not be visible in an MRI slice throughout the navigation. Although we have demonstrated a potential tracking method with alternating slice positions in our previous work²²these solutions would still be feasible for a limited workspace. 2D MR projection images with two orthogonal planes and fast 3D imaging methods might provide real-time tracking alternatives in the future; however, the increased imaging speed might be limited by the risk of tissue heating due to the specific absorption rate (SAR) of RF pulses. Additionally, the acoustic noise generated by rapidly switching gradient coils during fast MRI would also be a significant limitation for the use of faster imaging sequences during operations. Meanwhile, proposed NMR-based tracking methods do not cause heating since they do not deposit RF energy in the patient and can be used without generating any acoustic noise, as discussed in the Discussion section.

Secondly, MR image-based tracking is limited by certain conditions. MRI can only show objects in a water or tissue environment; in air, MR images do not provide meaningful information. Therefore, we cannot track objects that are fully or partly air-filled parts of the body, such as the GI tract. Moreover, we cannot track objects that approach from outside the patient and later penetrate the skin, such as needles and biopsy probes. MR image-based tracking would require auxiliary tracking methods to track objects until they reach the tissue. In contrast, NMR-based tracking can be performed in air, as demonstrated in our experiments.

To highlight the additional unique advantages of NMR-based tracking, we include the following in the Discussion section.

Additionally, the NMR magnetic sensing technology addresses certain drawbacks of MR image-based tracking. Despite continuous improvements in MRI speed, real-time 3D tracking with passive fiducial markers, such as magnetic ²², ¹⁹F 23, and RF coils²⁴, and active MRI pickup coils ²⁵, remains limited by slice thickness ²² and RF-induced tissue heating. In contrast, NMR sensors provide continuous 3D tracking without requiring the deposition of RF energy (see Supplementary Information), enabling continuous navigation with no risk of heating. Furthermore, NMR magnetic sensing eliminates acoustic noise, which in fast MRI-based tracking can exceed safe exposure limits ^{3,5} and hinder communication between clinicians during interventional MRI operations. Because NMR does not rely on MR gradients (figure S23, Movie S8), it is inherently silent, creating a safer acoustic environment for both patients and clinicians. Moreover, NMR tracking also functions in air-filled cavities, where most MRI-based tracking methods, except ¹⁹F tracking ²³, fail due to the absence of ¹H of the surrounding water or tissue, allowing for the uninterrupted tracking of tools such as needles during insertion from outside the skin.

We envision that the NMR magnetic sensing could also be used in conjunction with 2D MRI and recently emerging MRI-powered magnetic actuation methods, by using an alternating MR sequence similar to ⁵. We can perform high-speed magnetic tracking with NMR simultaneously during slower MR image acquisition, which is safe in terms of RF heating and acoustic noise, and we can also integrate magnetic actuation at the same time. Furthermore, while clinical use of high-field MRI scanners is becoming popular ²⁶, the integration of NMR magnetic sensor arrays into lower field MRI scanners such as 1.5 T/3 T and emerging ultra-low-field MRI scanners could increase the affordability of magnetically guided minimally invasive operations in the future ²⁷;

Comment 2.14) The paper notes that this method is unsuitable for patients with ferromagnetic implants and faces challenges in environments with dynamic tissues (e.g., the heart or nerves) due to background signal variations. These limitations may hinder the adoption of this technology in complex clinical scenarios. The authors are encouraged to discuss potential solutions to address these limitations.

Response: We thank the reviewer for the comment. Like other MR technologies, NMR magnetic tracking can be used for patients with ferromagnetic implants, as these patients may not be able to enter the MRI scanner due to the risk of the magnetic gradient fields from the MRI scanner dislocating the implant. On the other hand, the dynamic tissue-related background signal variation issues could be addressed in the future using an improved prediction model for the motion of the tracker and also the dynamic variation of the background. We have added a brief discussion on a potential method to address this issue.

The second limitation is the difficulty in tracking magnetic trackers from near-moving tissues. While the magnetization calibration enables tracking near static tissues by subtracting the diamagnetic background, tracking near moving tissues, such as the chest during breathing motion, is challenging due to changing background signals²⁰. This issue could be addressed in the future by incorporating the actuation dynamics into the particle filter rather than using a static model, and modifying the proposed background field estimation using basis functions that approximate the dynamic component of the background³⁰, while utilizing more NMR magnetic sensors for increased measurement redundancy.

In summary, this paper introduces an innovative NMR-based wireless magnetic tracking technology with advantages in sensitivity, miniaturization, and MRI compatibility. While the research shows great potential, some areas need further improvement, such as optimizing sensor layouts, addressing MRI interference, refining models for flexible trackers, and providing more experimental evidence. Additionally, comparisons with electromagnetic tracking and discussions on clinical limitations and applications would enhance the study's impact and practical relevance.

Response: We thank the reviewer for the kind comments.